# Exploiting Hidden Symmetry to Improve Objective Perturbation for DP Linear Learners with a Nonsmooth L1-norm

**Du Chen**[*]
Nanyang Business School
Nanyang Technological University
Singapore, 639798
`chen1443@e.ntu.edu.sg`

**Geoffrey A. Chua**
Nanyang Business School
Nanyang Technological University
Singapore, 639798
`gbachua@ntu.edu.sg`

## Abstract

Objective Perturbation (OP) is a classic approach to differentially private (DP) convex optimization with smooth loss functions but is less understood for nonsmooth cases. In this work, we study how to apply OP to DP linear learners under loss functions with a hidden or implicit $\ell_1$-norm structure, such as $\max\{0, x\}$ as a motivating example. We propose to first smooth out the implicit $\ell_1$-norm by convolution, and then invoke standard OP. Convolution has many advantages that distinguish itself from Moreau Envelope, such as approximating from above and a higher degree of hyperparameters. These advantages, in conjunction with the symmetry of $\ell_1$-norm, result in tighter pointwise approximation, which further facilitates tighter analysis of generalization risks by using pointwise bounds. Under mild assumptions on groundtruth distributions, the proposed OP-based algorithm is found to be rate-optimal, and can achieve the excess generalization risk $\mathcal{O}\left(\frac{1}{\sqrt{n}} + \frac{\sqrt{d \ln(1/\delta)}}{n\varepsilon}\right)$. Experiments demonstrate the competitive performance of the proposed method to Noisy-SGD.

## 1 Introduction

Differentially private convex optimization is one of the most crucial tools in private data analysis, which seeks a well-performing output from an optimization problem so that the output is also insensitive to the presence or absence of an individual in a dataset. In the past decade or more, numerous works have together formed a good understanding of DP convex optimization, such as Chaudhuri et al. (2011); Kifer et al. (2012); Bassily et al. (2014; 2019; 2020), to name a few. While Noisy-SGD (Abadi et al., 2016) has outstripped almost all other approaches, findings of privacy leakage through Noisy-SGD's hyperparameter tuning (Papernot & Steinke, 2022; Mohapatra et al., 2022) have motivated recent revisits (Redberg et al., 2023; Agarwal et al., 2023) to another classical and competitive method, Objective Perturbation (Kifer et al., 2012, OP).

OP follows a completely different design philosophy: it injects noise into the loss function of an empirical risk minimization (ERM) problem and then solves the noisy ERM, unlike Noisy-SGD that injects noise into gradients. Often, OP only requires light to no hyperparameter tuning. Moreover, because the noisy ERM can be solved by any optimizer whose choice is independent of the problem at hand, OP often returns high-quality (empirical) minimizers, regardless of the sample size. To further improve its practicability, the recent work (Redberg et al., 2023) has tightened privacy accounting of OP by employing privacy profiles, making it competitive or even better in performance than honest Noisy-SGD ("honest" means privacy in hyperparameter tunings is correctly tracked). Some other works also found OP appealing for certain tasks, such as private logistic regression (Iyengar et al., 2019), binary classification (Neel et al., 2020), quantile regression (Chen & Chua, 2023), and online convex optimization (Agarwal et al., 2023). As a subroutine, OP also found its

---

[*]Corresponding Author

place in personalized pricing (Chen et al., 2022) that fosters analysis of statistical properties of some estimators.

However, compared to Noisy-SGD that can be applied to both nonsmooth and smooth problems, OP is often criticized for being too restrictive as its performance is known to be optimal only when being applied to problems with smooth loss functions. Intuitively, this is because minimizers to nonsmooth functions are very unstable. To hide privacy contained in such unstable minimizers, we should either inject a high-variance noise or stabilize the minimizer first, both of which might engender nontrivial accuracy loss. This technical issue prevents a rich class of problems from enjoying the advantages of OP, ranging from simple problems with $\|\cdot\|_1$-regularizer to neural networks with ReLU activation functions. As a motivating example demonstrated by Bassily et al. (2014), minimizing a simple though basic nonsmooth function $\widehat{\mathcal{L}}(\boldsymbol{\theta}) := \sum_{i=1}^n \max\{0, y_i - \boldsymbol{\theta}^\top \boldsymbol{x}_i\}$ by OP is found to be challenging.

While the nonsmoothness issue of OP has been observed for a decade, it has not been well addressed yet. One promising remedy is to smooth out the original nonsmooth function to get a smooth approximation, and then apply standard OP to the smoothed approximation function. However, a downside of this idea is the additional approximation error introduced by the smoothing step. Early on, Bassily et al. (2014) has noticed severe consequences of the additional error: the convergence rates of OP is no longer optimal. They attempted to resolve the issue with convolution smoothing, but ended with concluding that "straightforward smoothing does not yield optimal algorithms". More recently, some works (Kulkarni et al., 2021; Chen & Chua, 2023) designed more intricate OP-based algorithms and developed involved analysis. But they either failed to obtain optimal convergence rates or achieved it only under strong regularity conditions. Approaches that do not include a smoothing step has also been explored by Neel et al. (2020), but they left the convergence rate-optimality as an unanswered question.

Observing the strengths of OP and the open problem for nonsmooth cases, we are interested in applying OP to nonsmooth convex problems. Specifically, we focus our discussion mainly on a common class of nonsmooth problems, where the loss function $f(\boldsymbol{\theta}; \boldsymbol{z})$ can be written as a sum of an $\ell_1$-norm function $\|A(\boldsymbol{z})\boldsymbol{\theta}\|_1$ and a well-behaved convex smooth function $h(\boldsymbol{\theta}^\top \boldsymbol{z})$, where $\boldsymbol{\theta}$ is the variable to optimize and $\boldsymbol{z}$ is the data point, and $A(\cdot), h(\cdot)$ are some known functions. In other words, we assume that the nonsmoothness issue in the loss function $f(\boldsymbol{\theta}; \boldsymbol{z}) := \|A(\boldsymbol{z})\boldsymbol{\theta}\|_1 + h(\boldsymbol{\theta}^\top \boldsymbol{z})$ is rooted in the $\ell_1$ norm.

This structure naturally covers problems with a (possibly grouped) $\ell_1$-regularizer $\|\cdot\|_1$ with a linear transformation of $A(\cdot)$. Moreover, many loss functions with widely accepted formulations not following the structure can actually be reformulated into this form; for instance, positive part operator $x^+ = |x|/2 + x/2$, pinball loss $rx^+ + (1-r)(-x)^+ = |x|/2 + (r-1/2)x$, $\tau$-soft-thresholding $(|x| - \tau)^+ = \left\| \begin{smallmatrix} (x+\tau)/2 \\ (x-\tau)/2 \end{smallmatrix} \right\|_1 - \tau$, etc. This is precisely why we call it an implicit or hidden $\ell_1$ structure throughout this paper, starting from its title. Surprisingly, some special functions that do not follow the considered structure are also found to enjoy the advantages developed in this work, see Section 4.3 for extensions.

The key step of our approach is to apply convolution smoothing (Hirschman & Widder, 2012) to the $\ell_1$-norm function only. While it seems a straightforward idea and a minor refinement, the benefit turns out to be significant. The reason is that the symmetry of the $\ell_1$ function can be utilized to identify a smaller set over which an integral is calculated to characterize pointwise approximation errors. With proper kernels, the errors could be exponentially small, in contrast to linearly small errors used in the literature. The exponentially small approximation error is negligible and thus does not harm convergence rates. Moreover, the smoothing method chosen is convolution smoothing, rather than the most common method, Moreau Envelope (Parikh et al., 2014). Though convolution is a classical method, its advantages such as analytic convenience and a higher degree of hyperparameter are not often noticed and exploited in DP literature. It turns out that these overlooked advantages are crucial to the improved performance. We compare both methods in Section 4.4.

**Our contributions.** Our first contribution includes the adoption of convolution for nonsmooth DP convex optimization problems and demonstrating its advantages in performance improvements. While we are not the first to employ convolution to address nonsmoothness issues, our analysis provides new insights into its role. Along with this development, we also make a thorough comparison to Moreau Envelope. The second contribution is the improved performance analysis of convergence

rates of the proposed OP-based algorithm for nonsmooth DP stochastic convex optimization (DP-SCO) with an implicit $\ell_1$ structure. Under mild assumptions, the proposed algorithm can achieve optimal rates of DP-SCO in a Euclidean space. Last, we run simulations to demonstrate the benefits of convolution and OP. Specifically, we observe a comparable performance to Noisy-SGD, and OP even performs better in high-privacy regimes.

**Related works.** Objective Perturbation (Kifer et al., 2012, OP) is a classical tool for DP-ERM (Bassily et al., 2014) and DP-SCO (Bassily et al., 2019). OP outputs a minimizer of a perturbed loss function, which substantially differs from iterative algorithms, such as Noisy-SGD (Abadi et al., 2016). For smooth Generalized Linear Models, OP is known to be rate optimal, i.e. its excess generalization risk is $\mathcal{O}(\frac{1}{\sqrt{n}} + \frac{\sqrt{d \ln (1/\delta)}}{n\varepsilon})$ (Bassily et al., 2021). To improve OP's practicability, under smoothness assumption, Iyengar et al. (2019) extended OP to allow it to return approximate minimizers, and Redberg et al. (2023) tightened privacy accounting. However, much less is known about nonsmooth cases. Without smoothness, Neel et al. (2020) proposed an OP algorithm paired with an additional output perturbation step, but whether their algorithm is optimal is unclear. Chen & Chua (2023) considered a special case of quantile regression; however, their result lacks generality. Our work is also closely related to another stream of work that applies convolution to address nonsmoothness issues in DP convex optimization (Feldman et al., 2018; Kulkarni et al., 2021; Wang et al., 2021; Carmon et al., 2023). A common feature of these works is that they all apply convolution before feeding loss functions into a standard OP. We follow this idea to develop our algorithm.

## 2 PRELIMINARIES

**Definition 2.1** (Differential Privacy). A randomized algorithm $\mathcal{A} : \mathcal{Z}^n \to \Theta$ is $(\varepsilon, \delta)$-differentially private if, for any pair of neighboring datasets $\mathcal{D} \sim \mathcal{D}'$ that differ in one data point, and for any subset $\mathcal{S} \subseteq \Theta$, $\Pr\left[\mathcal{A}(\mathcal{D}) \in \mathcal{S}\right] \leq e^\varepsilon \cdot \Pr\left[\mathcal{A}(\mathcal{D}') \in \mathcal{S}\right] + \delta$.

**Definition 2.2** ($\beta$-smoothness). Let $\beta \geq 0$. A function $f : \Theta \to \mathbb{R}$ is $\beta$-smooth (w.r.t. $\|\cdot\|_p$) over a set $\Theta$ if for every $\boldsymbol{\theta}_1, \boldsymbol{\theta}_2 \in \Theta$, $\|\nabla f(\boldsymbol{\theta}_1) - \nabla f(\boldsymbol{\theta}_2)\|_q \leq \beta \|\boldsymbol{\theta}_1 - \boldsymbol{\theta}_2\|_p$, where $p, q$ are conjugate indices such that $1/p + 1/q = 1$. If the only admissible value of $\beta$ is $\infty$, we say $f$ is nonsmooth.

**Notation.** We use $\mathcal{B}(R) := \{\boldsymbol{\theta} \in \mathbb{R}^d : \|\boldsymbol{\theta}\|_2 \leq R\}$ to denote the Euclidean ball with radius $R > 0$ around the origin, and $\|\cdot\|_2$ to denote Euclidean norm. Data space is $\mathcal{Z}$, and datapoints in a dataset $\mathcal{D} := \{\boldsymbol{z}_i\}_{i=1}^n$ are i.i.d. drawn from an unknown distribution $\mathbb{P}$ supported on $\mathcal{Z}$. The empirical risk of any $\boldsymbol{\theta} \in \Theta \subseteq \mathbb{R}^d$ under loss function $f$ and dataset $\mathcal{D}$ is denoted by $\widehat{F}(\boldsymbol{\theta}; \mathcal{D}) := \frac{1}{n} \sum_{i=1}^n f(\boldsymbol{\theta}; \boldsymbol{z}_i)$, and the generalization risk of $\boldsymbol{\theta}$ under distribution $\mathbb{P}$ is denoted by $F(\boldsymbol{\theta}; \mathbb{P}) := \mathbb{E}_{\boldsymbol{z} \sim \mathbb{P}}\left[f(\boldsymbol{\theta}; \boldsymbol{z})\right]$. Shorthand $\widehat{F}(\boldsymbol{\theta})$ and $F(\boldsymbol{\theta})$ are used when the dependence is clear from the context. The excess generalization risk of algorithm $\mathcal{A}$ under distribution $\mathbb{P}$ is thus denoted as $\mathcal{R}(\mathcal{A}; \mathbb{P}) := \mathbb{E}_{\mathcal{D} \sim \mathbb{P}^n, \mathcal{A}}\left[F(\widehat{\boldsymbol{\theta}}^{\mathcal{A}})\right] - F(\boldsymbol{\theta}^*)$ where $\boldsymbol{\theta}^* := \arg\min_{\boldsymbol{\theta} \in \mathbb{R}^d} F(\boldsymbol{\theta})$.

In this work, our ultimate goal is to design an OP-based algorithm $\mathcal{A}$ to find $\widehat{\boldsymbol{\theta}}^{\mathcal{A}}$ that can achieve rate-optimal performance in terms of excess generalization risk $\mathcal{R}(\mathcal{A}; \mathbb{P}) := \mathbb{E}_{\mathcal{D} \sim \mathbb{P}^n, \mathcal{A}}\left[F(\widehat{\boldsymbol{\theta}}^{\mathcal{A}})\right] - F(\boldsymbol{\theta}^*)$ for nonsmooth functions satisfying a specific structure in Assumption 2.3.

**Assumption 2.3** (Nonsmooth models with an $\ell_1$ structure).

1. **(Implicit $\ell_1$ structure)** Loss function $f(\boldsymbol{\theta}; \boldsymbol{z}) : \mathbb{R}^d \times \mathbb{R}^d \to \mathbb{R}$ has an implicit $\ell_1$-norm structure and can be written as $f(\boldsymbol{\theta}; \boldsymbol{z}) := \|A(\boldsymbol{z})\boldsymbol{\theta}\|_1 + h(\boldsymbol{z}^\top \boldsymbol{\theta})$ for some known function $h(\cdot) : \mathbb{R} \to \mathbb{R}$ and function $A(\cdot) : \mathbb{R}^d \to \mathbb{R}^{m \times d}$, where $m \leq d$ is independent of $d$.

2. **(Well-behaved $h(\cdot)$)** Function $h(\cdot)$ is convex and $\beta_h$-smooth in $\boldsymbol{\theta}$ (w.r.t. $\|\cdot\|_2$). As a scalar function, its derivative is uniformly upper bounded by $L_h$.

3. **(Boundedness)** Let $\boldsymbol{\theta}^* := \arg\min_{\boldsymbol{\theta} \in \mathbb{R}^d} F(\boldsymbol{\theta})$. We asume $\boldsymbol{\theta}^* \in \mathcal{B}(R)$. Data space $\mathcal{Z} := \mathcal{B}(D) \subseteq \mathbb{R}^d$ is a Euclidean ball. Further assume the 2-norm of matrix $A(\cdot)$ is uniformly upper bounded, i.e. $\sup_{\boldsymbol{z} \in \mathcal{Z}} \|A(\boldsymbol{z})\|_2 \leq \overline{A}$.

The first assumption makes our discussion focus on a specific class of nonsmooth functions where the nonsmoothness comes from the implicit $\ell_1$-norm. Despite the structural assumption, the con-

sidered model still covers a rich set of interesting problems. For instance, the motivating example $\max\{0, x\} = (|x| + x)/2$ admits a reformulation with $A(\boldsymbol{z}) := \boldsymbol{z}^\top/2$ and $h(x) := x/2$, where $x$ is the residual derived from $\boldsymbol{z}^\top\boldsymbol{\theta}$. Similar reformulation applies to pinball loss: $\forall r \in (0, 1)$, we have $rx^+ + (1 - r)(-x)^+ = |x|/2 + (r - 1/2)x$. Another illustrative example is when $A(\cdot)$ is independent of datapoint $\boldsymbol{z}$; then the original model becomes an $\ell_1$-regularized GLM. If we further have $h \equiv 0$, then the problem becomes a model for finding high-dimensional quantiles. Given all these examples, it should be clear that our assumption of the model structure is not very restrictive. The second assumption ensures $h(\cdot)$ is well behaved, and the third assumption on boundedness is common and appears frequently in DP literature.

## 3 THE ALGORITHM

We propose the algorithm Convolution-then-Objective Perturbation (C-OP), which is formally given below in Algorithm 1. The algorithm is built upon classical OP by wrapping it with an additional convolution smoothing step (1). The smoothed function is then fed into classical OP, i.e. Step 4, which returns minimizer $\widehat{\boldsymbol{\theta}}^{\mathcal{A}}$. Both privacy and performance guarantees highly depend on the convolution step. We thus give a brief introduction to convolution smoothing first.

---

**Algorithm 1** Convolution then Objective Perturbation (C-OP), $\mathcal{A}_{\text{C-OP}}$

---

**Input:** Private dataset $\mathcal{D} := \{\boldsymbol{z}_i\}_{i=1}^n$; privacy parameters $(\varepsilon, \delta)$; noise variance $\sigma^2$; nonsmooth loss function $f(\boldsymbol{\theta}; \boldsymbol{z}) = \|A(\boldsymbol{z})\boldsymbol{\theta}\|_1 + h(\boldsymbol{z}^\top\boldsymbol{\theta})$ that satisfies Assumption 2.3; Constant $C := \sqrt{m\overline{A}^2 + D^2 L_h^2}$; any random variable $\mathbf{k}$ whose pdf (i.e. kernel) is given in the first column in Table 1, and bandwidth parameter $\mu > 0$.

1: For a given $\lambda$, find $\mu$ such that $\lambda = \frac{(\beta_\mu + \beta_h)(m+1)}{n\varepsilon}$, where $\beta_\mu$ is given in Table 1.
2: Get smooth approximation by convolution,

$$f_\mu(\boldsymbol{\theta}; \boldsymbol{z}) = \mathbb{E}_{\mathbf{k}}\left[\|A(\boldsymbol{z})\boldsymbol{\theta} + \mu\mathbf{k}\|_1\right] + h(\boldsymbol{z}^\top\boldsymbol{\theta}) \tag{1}$$

3: Draw a Gaussian noise vector $\boldsymbol{b} \sim \mathcal{N}(0, \sigma^2 \mathbf{I}_{d\times d})$
4: $\widehat{\boldsymbol{\theta}}^{\mathcal{A}} \leftarrow \arg\min_{\boldsymbol{\theta}\in\mathbb{R}^d} \frac{1}{n}\sum_{i=1}^n f_\mu(\boldsymbol{\theta}; \boldsymbol{z}_i) + \lambda\|\boldsymbol{\theta}\|_2^2 + \frac{\langle \boldsymbol{b}, \boldsymbol{\theta}\rangle}{n}$
5: **Return:** $\widehat{\boldsymbol{\theta}}^{\mathcal{A}}$

---

### 3.1 CONVOLUTION SMOOTHING

Convolution smoothing Hirschman & Widder (2012) is an operation on function $g : \mathbb{R}^m \to \mathbb{R}_+$ and kernel $k : \mathbb{R}^m \to \mathbb{R}_+$ that produces a smooth approximation $g_\mu$ of $g$. The kernel function $k$ should meet some regularity Conditions A.1 in the Appendix. In the main text, we focus on three common kernels listed in Table 1. Each kernel in Table 1 defines a probability density function (pdf). Intuitively, the approximated function value $g_\mu(\boldsymbol{x}) := \mathbb{E}_{\mathbf{k}}[g(\boldsymbol{x} + \mu\mathbf{k})]$ is a weighted average over

Table 1: Kernels and properties of smooth approximation (Lemma 3.1)

| | | properties of $g_\mu(\boldsymbol{x}) := \mathbb{E}_{\mathbf{k}}[g(\boldsymbol{x} + \mu\mathbf{k})]$ | | | |
|---|---|---|---|---|---|
| Kernels | $k(\boldsymbol{v})$ | Lipschitz $L_\mu$ | smoothness $\beta_\mu$ | uniform gap $\sup_{\boldsymbol{x}}(g_\mu - g)(\boldsymbol{x})$ | pointwise gap $(g_\mu - g)(\boldsymbol{x})$ |
| Gaussian | $e^{-\frac{\|\boldsymbol{v}\|_2^2}{2}}/(\sqrt{2\pi})^m$ | $L$ | $L/\mu$ | $L\mu\kappa_p$ | $L\mu\int_{\mathcal{V}_\mu(\boldsymbol{x})}\|\boldsymbol{v}\|_p k(\boldsymbol{v})\,d\boldsymbol{v}$ |
| Exponential | $e^{-\|\boldsymbol{v}\|_2}/\mathfrak{n}$ | $L$ | $\sqrt{6}L/\mu$ | $L\mu\kappa_p$ | same as above |
| Laplacian | $e^{-\|\boldsymbol{v}\|_1}/2^m$ | $L$ | $\begin{cases}\sqrt{6}L/\mu, \text{ if } p = 1 \\ \sqrt{6m}L/\mu, \text{ o.w.}\end{cases}$ | $L\mu\kappa_p$ | same as above |

*Notes.* Properties of $g_\mu$ under Gaussian kernel is known in the literature (Duchi et al., 2012); we derive properties for other kernels. The set $\mathcal{V}_\mu(\boldsymbol{x})$ in the last column is defined around Lemma 4.1. The normalizer $\mathfrak{n}$ for exponential kernel is $\mathfrak{n} = \Gamma(m/2)/(2\pi^{m/2}\Gamma(m))$ with Gamma function $\Gamma(\cdot)$.

its neighbors, and the weights are controlled by kernel $k$ and bandwidth parameter $\mu > 0$. Properties of the approximation function $g_\mu$ for general Lipschitz continuous function $g$ are given below.

**Lemma 3.1** (Properties of $g_\mu$). *Let $g : \mathbb{R}^m \to \mathbb{R}_+$ be a closed, proper, convex, and $L$-Lipschitz continuous (w.r.t $\|\cdot\|_p$, $p \in [1,2]$) loss function. Let $k : \mathbb{R}^m \to \mathbb{R}_+$ be any kernel function in Table 1; denote $\kappa_p := \mathbb{E}_{\mathbf{k}}\left[\|\mathbf{k}\|_p\right]$. Then, the convolution smoothing $g_\mu$ possesses following properties:*

1. *$g_\mu$ is convex, $L_\mu$-Lipschitz and $\beta_\mu$-smooth w.r.t. $\|\cdot\|_p$ (see Table 1 for values of $L_\mu$ and $\beta_\mu$);*

2. *$g_\mu$ is differentiable with gradient $\nabla g_\mu(\boldsymbol{x}) = \mathbb{E}_{\mathbf{k}}\left[\nabla g(\boldsymbol{x} + \mu \mathbf{k})\right], \forall \boldsymbol{x}$;*

3. *Approximation error satisfies the inequality $g(\boldsymbol{x}) \leq g_\mu(\boldsymbol{x}) \leq g(\boldsymbol{x}) + L\mu\kappa_p, \forall \boldsymbol{x}$;*

4. *$g_\mu(\boldsymbol{x}) = \int_{\boldsymbol{v} \in \mathbb{R}^m} \left[\frac{g(\boldsymbol{x}+\mu\boldsymbol{v})+g(\boldsymbol{x}-\mu\boldsymbol{v})}{2}\right] k(\boldsymbol{v})\, d\boldsymbol{v}.$*

These properties hold for general convex and Lipschitz function $g_\mu$. Throughout this paper, we will use $g = g^{\ell_1} := \|\cdot\|_1$ frequently.

## 3.2 Preliminary results

It can be shown that, with a well-calibrated variance $\sigma^2$, the algorithm C-OP is $(\varepsilon, \delta)$-DP.

**Theorem 3.2** (Privacy Guarantee). *Suppose Assumption 2.3 holds. The algorithm $\mathcal{A}_{\text{C-OP}}$ is $(\varepsilon, \delta)$-DP, if $\sigma^2 \geq \frac{C^2 \cdot (8\ln(1/\delta) + 8\varepsilon)}{\varepsilon^2}$ where $C := \sqrt{m\overline{A}^2 + L_h^2 D^2}$.*

Because of the matrix $A(\boldsymbol{z})$ in Assumption 2.3, our model does not have the same model structure for which the privacy accounting bug was fixed by Redberg et al. (2023) and Agarwal et al. (2023). Thus, we provide a detailed proof in the Appendix. The proof follows a similar idea in Agarwal et al. (2023) but uses bounded $A(\cdot)$ to control the privacy loss random variable's tail behavior.

Now, we move on to analyze the performance of C-OP. A crucial observation is the third part of Lemma 3.1, which implies $g_\mu^{\ell_1}(\boldsymbol{x}) := \mathbb{E}_{\mathbf{k}}[\|\boldsymbol{x} + \mu \mathbf{k}\|_1]$ approximates original function $g^{\ell_1}(\boldsymbol{x}) := \|\boldsymbol{x}\|_1$ from above. Therefore, if we apply convolution to the $\ell_1$ part of loss function $f$, then $f_\mu$ given by (1) is a pointwise upper bound for $f$. This facilitates a new decomposition of the excess generalization risk $\mathcal{R}(\mathcal{A}; \mathbb{P}) := \mathbb{E}_{\mathcal{D} \sim \mathbb{P}^n, \mathcal{A}}\left[F(\widehat{\boldsymbol{\theta}}^{\mathcal{A}})\right] - F(\boldsymbol{\theta}^*)$, as shown below:

$$
\begin{aligned}
\mathcal{R}(\mathcal{A}; \mathbb{P}) &= \mathbb{E}_{\mathcal{D},\mathcal{A}}\left[F(\widehat{\boldsymbol{\theta}}^{\mathcal{A}}) - \widehat{F}(\widehat{\boldsymbol{\theta}}^{\mathcal{A}})\right] + \mathbb{E}_{\mathcal{D},\mathcal{A}}\left[\widehat{F}(\widehat{\boldsymbol{\theta}}^{\mathcal{A}}) - F(\boldsymbol{\theta}^*)\right] \\
&\leq \mathbb{E}_{\mathcal{D},\mathcal{A}}\left[F(\widehat{\boldsymbol{\theta}}^{\mathcal{A}}) - \widehat{F}(\widehat{\boldsymbol{\theta}}^{\mathcal{A}})\right] + \mathbb{E}_{\mathcal{D},\mathcal{A}}\left[\widehat{F}_\mu(\widehat{\boldsymbol{\theta}}^{\mathcal{A}}) - F(\boldsymbol{\theta}^*)\right] \\
&= \mathbb{E}_{\mathcal{D},\mathcal{A}}\left[F(\widehat{\boldsymbol{\theta}}^{\mathcal{A}}) - \widehat{F}(\widehat{\boldsymbol{\theta}}^{\mathcal{A}})\right] + \mathbb{E}_{\mathcal{D},\mathcal{A}}\left[\widehat{F}_\mu(\widehat{\boldsymbol{\theta}}^{\mathcal{A}}) - \widehat{F}_\mu(\boldsymbol{\theta}^*)\right] + [F_\mu(\boldsymbol{\theta}^*) - F(\boldsymbol{\theta}^*)]. \quad (2)
\end{aligned}
$$

In the first line, we insert terms $\widehat{F}(\widehat{\boldsymbol{\theta}}^{\mathcal{A}})$; in the second line, we use the pointwise upper bound $f_\mu \geq f$; in the third line, we insert $\widehat{F}_\mu(\boldsymbol{\theta}^*)$. Essentially, the new risk upper bound (2) consists of three parts. The first part is a sampling error that can be controlled through uniform stability analysis; the second part is an empirical risk that can be controlled through risk analysis. With these observations, we can get a preliminary result of C-OP's performance.

**Lemma 3.3** (C-OP Performance; Preliminary). *Suppose Assumption 2.3 holds. If we set the regularizer coefficient $\lambda = \sqrt{4C^2/n + d\sigma^2/n^2}/R$ and use $\sigma^2$ suggested by Theorem 3.2, then*

$$
\mathcal{R}(\mathcal{A}; \mathbb{P}) \leq 4\sqrt{2} C R \cdot \left(\frac{1}{\sqrt{n}} + \frac{\sqrt{d\ln(1/\delta)}}{n\varepsilon}\right) + [F_\mu(\boldsymbol{\theta}^*) - F(\boldsymbol{\theta}^*)].
$$

The preliminary performance bound is remarkable, as it suggests that the additional approximation error depends only on the approximation quality at optimal $\boldsymbol{\theta}^*$. Intuitively, this comes from using pointwise approximation upper bound to tightly characterize the approximation error, which is unique to convolution smoothing. Instead, many other smoothing methods, such as Moreau Envelope, do not allow this tighter characterization, roughly because Moreau approximates from below

(see details in Section 4.4). With lemma 3.3, it remains to control the population-level approximation error at $\boldsymbol{\theta}^*$. We do so in the next section by exploiting the $\ell_1$ structure of assumed models.

Before proceeding, we want to highlight an immediate result on the choice of $\mu$ from Lemma 3.3 and Theorem 3.2. According to Step 1 of Algorithm 1, the value of $\mu$ should satisfy $\lambda = \frac{(\beta_\mu + \beta_h)(m+1)}{n\varepsilon}$. Because $\beta_\mu \asymp \frac{1}{\mu}$, $\lambda \asymp \sqrt{1/n + d\sigma^2/n^2}$, and $\sigma^2 \asymp \ln(1/\delta)/\varepsilon^2$, we know $\mu \asymp \frac{m}{\sqrt{n\varepsilon^2 + 2d(\ln(1/\delta)+\varepsilon)} - \beta_h}$. Therefore, roughly speaking, when sample size $n \to \infty$, the value of $\mu$ decreases. The fact that $\mu$ is decreasing is a desired feature, leading to $F_\mu(\boldsymbol{\theta}^*) - F(\boldsymbol{\theta}^*) \to 0$ (see next section for details).

# 4 IMPROVED APPROXIMATION AND OPTIMAL RATES

## 4.1 EXPLOIT SYMMETRY OF $\ell_1$-NORM TO OBTAIN TIGHTER APPROXIMATION ERRORS

We know from part four of Lemma 3.1 that $g_\mu(\boldsymbol{x})$ is in fact a convex combination of $g(\boldsymbol{x} + \mu\boldsymbol{v})$ and $g(\boldsymbol{x} - \mu\boldsymbol{v})$; thus the approximation error at $\boldsymbol{x}$ admits a closed form

$$g_\mu(\boldsymbol{x}) - g(\boldsymbol{x}) = \int_{\boldsymbol{v}\in\mathbb{R}^m} \left[ \frac{g(\boldsymbol{x}+\mu\boldsymbol{v}) + g(\boldsymbol{x}-\mu\boldsymbol{v})}{2} - g(\boldsymbol{x}) \right] k(\boldsymbol{v})\, d\boldsymbol{v}, \forall \boldsymbol{x}. \tag{3}$$

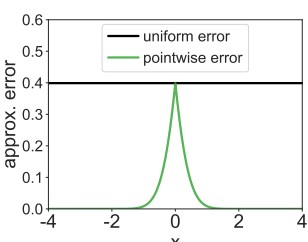

However, most existing works use the uniform upper bound $L\mu\kappa_p$ in part three of Lemma 3.1 for convergence analysis, which is obviously too conservative and significantly overestimates actual approximation errors. The example of $g(x) = |x|$ in Figure 1 demonstrates the huge overestimate: outside the interval $[-1, 1]$ roughly, there is no approximation error by convolution (green curve in Figure 1), but uniform bound (in black) says the error is nontrivial.

Figure 1: approximation error. $\mu = 0.5$, Gaussian kernel

To understand this phenomenon analytically, we first notice that the pointwise error (3) is calculated from an integral on the entire space. However, it actually suffices to integrate over a smaller set $\mathcal{V}_\mu(\boldsymbol{x})$, where the integrand $\frac{g(\boldsymbol{x}+\mu\boldsymbol{v})+g(\boldsymbol{x}-\mu\boldsymbol{v})}{2} - g(\boldsymbol{x})$ is *strictly* positive:

$$\mathcal{V}_\mu(\boldsymbol{x}) := \{\boldsymbol{v} \in \mathbb{R}^m : g(\boldsymbol{x}+\mu\boldsymbol{v}) + g(\boldsymbol{x}-\mu\boldsymbol{v}) > 2g(\boldsymbol{x})\}. \tag{4}$$

Although the integrand is always nonnegative by convexity, under $\ell_1$-norm function, the set $\mathcal{V}_\mu(\boldsymbol{x})$ is in fact a much smaller set than the entire space, thanks to the symmetry of $\ell_1$-norm function. For clarity, we denote this set under $\ell_1$-norm function by $\mathcal{V}_\mu^{\ell_1}(\boldsymbol{x}) := \{\boldsymbol{v} \in \mathbb{R}^m : \|\boldsymbol{x}+\mu\boldsymbol{v}\|_1 + \|\boldsymbol{x}-\mu\boldsymbol{v}\|_1 > 2\|\boldsymbol{x}\|_1\}$. It has a closed-form expression as shown below.

**Lemma 4.1** (Smaller Domain of Integration). *if $g^{\ell_1} : \boldsymbol{x} \mapsto \|\boldsymbol{x}\|_1$ is the $\ell_1$-norm function, then the set $\mathcal{V}_\mu^{\ell_1}(\boldsymbol{x})$ on which the integrand $\frac{\|\boldsymbol{x}+\mu\boldsymbol{v}\|_1 + \|\boldsymbol{x}-\mu\boldsymbol{v}\|_1}{2} - \|\boldsymbol{x}\|_1$ is strictly positive has a closed-form expression $\mathcal{V}_\mu^{\ell_1}(\boldsymbol{x}) := \{\boldsymbol{v} \in \mathbb{R}^m : |\boldsymbol{v}| > |\boldsymbol{x}|/\mu\}$, where $|\cdot|$ applies elementwise.*

Geometrically, the set $\mathcal{V}_\mu^{\ell_1}(\boldsymbol{x})$ is a hollow set with a rectangle around the origin being removed. Moreover, it can be shown that the set $\mathcal{V}_\mu$ defined in (4) shrinks with gradually decreasing $\mu \searrow 0$. This result holds for any convex function. But, specifically for $g^{\ell_1}$, this shrinkage is strict.

**Lemma 4.2** (Monotonicity of $\mathcal{V}_\mu$ in $\mu$). *If $g$ is convex but not linear, then for any given $\boldsymbol{x}$, the set $\mathcal{V}_\mu(\boldsymbol{x})$ is monotonically increasing in $\mu$ and satisfies, for any $0 < \mu_0 < \mu_1 < \infty$,*

$$\emptyset = \mathcal{V}_0(\boldsymbol{x}) \subseteq \mathcal{V}_{\mu_0}(\boldsymbol{x}) \subseteq \mathcal{V}_{\mu_1}(\boldsymbol{x}) \subseteq \mathcal{V}_\infty(\boldsymbol{x}) = \mathbb{R}^m\backslash\{\boldsymbol{0}\}, \quad \forall \boldsymbol{x}.$$

*Moreover, the inequality becomes strict under function $g^{\ell_1} := \|\cdot\|_1$, i.e.,*

$$\emptyset = \mathcal{V}_0^{\ell_1}(\boldsymbol{x}) \subset \mathcal{V}_{\mu_0}^{\ell_1}(\boldsymbol{x}) \subset \mathcal{V}_{\mu_1}^{\ell_1}(\boldsymbol{x}) \subset \mathcal{V}_\infty^{\ell_1}(\boldsymbol{x}) = \mathbb{R}^m\backslash\{\boldsymbol{0}\}, \quad \forall \boldsymbol{x} \in \partial g^{\ell_1},$$

*where $\partial g^{\ell_1} := \{\boldsymbol{x} \in \mathbb{R}^m : x_j \neq 0, \forall j = 1, \ldots, m\}$ is the set of differentiable points.*

This lemma allows us to further tighten the approximation errors. To ses this, we notice first that

$$g_\mu^{\ell_1}(\boldsymbol{x}) - g^{\ell_1}(\boldsymbol{x}) = \int_{\boldsymbol{v} \in \mathcal{V}_\mu^{\ell_1}(\boldsymbol{x})} \left[ \frac{\|\boldsymbol{x} + \mu\boldsymbol{v}\|_1 + \|\boldsymbol{x} - \mu\boldsymbol{v}\|_1}{2} - \|\boldsymbol{x}\|_1 \right] k(\boldsymbol{v}) \, d\boldsymbol{v} \le \mu \int_{\mathcal{V}_\mu^{\ell_1}(\boldsymbol{x})} \|\boldsymbol{v}\|_1 \, k(\boldsymbol{v}) \, d\boldsymbol{v} \,.$$

$$(5)$$

The upper bound in Eq.(5) is a product between the multiplicative factor $\mu$ and an integral over $\mathcal{V}_\mu^{\ell_1}$. Essentially, Lemma 4.2 states that when $\mu \searrow 0$, the set $\mathcal{V}_\mu^{\ell_1}$ tends to be an empty set. Therefore, both the factor $\mu$ and the integral term tend to 0 when $\mu \searrow 0$, indicating a much faster rate of their product than any of them. Moreover, the integral term decreases at a rate that heavily depends on the kernel function chosen. If we choose kernels from Table 1, then the integral term decreases exponentially fast due to the kernel functions' light tail. We formally characterize the finding below.

**Lemma 4.3** (Kernel-dependent Approx. Error). *Suppose $g^{\ell_1} : \boldsymbol{x} \mapsto \|\boldsymbol{x}\|_1$. If kernel function $k(\cdot)$ is Gaussian kernel, then $g_\mu^{\ell_1}(\boldsymbol{x}) - g^{\ell_1}(\boldsymbol{x}) \le \sqrt{2/\pi}\mu \cdot \sum_{j=1}^m \exp\left(-\frac{|x_j|^2}{2\mu^2}\right).$*

The left panel of Figure 2 justifies the finding: when $\mu \searrow 0$, the approximation error at either $x = 1$ or $x = 0.5$ decreases exponentially fast, whereas the uniform bound is only linear in $\mu$. This observation also supports our argument that using a uniform bound is too conservative for convergence analysis. Moreover, all observations naturally extend to high-dimensional cases, see the right panel of Figure 2. One remark about Lemmas 4.2 and 4.3 is that both results provide nontrivial improvements when $\boldsymbol{x} \notin \partial g^\ell$. Because the set $\partial g^{\ell_1}$ actually has Lebesgue measure zero, we can expect its impact to be negligible.

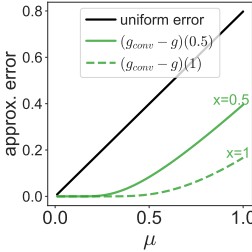
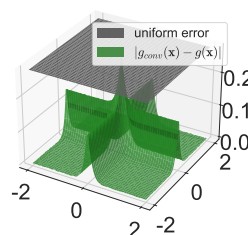

Figure 2: Left: approx. error v.s. $\mu$ under $g(x) = |x|$; convolution by Gaussian kernel. Right: approx. error under $g(\boldsymbol{x}) = \|\boldsymbol{x}\|_1$ in a 2-dim space.

## 4.2 OPTIMAL RATES OF C-OP

With developed tighter approximation characterization, we are ready to show optimal rates of C-OP under some distributions. By lemma 3.3, it suffices to show $F_\mu(\boldsymbol{\theta}^*) - F(\boldsymbol{\theta}^*)$ is dominated by $\mathcal{O}\left(\frac{1}{\sqrt{n}} + \frac{\sqrt{d\ln(1/\delta)}}{n\varepsilon}\right)$. Because approximation errors are roughly exponentially small (given Gaussian kernel is used), i.e. $\exp\left(-|A(\boldsymbol{z})\boldsymbol{\theta}^*|_j^2/\mu^2\right), \forall j \in [m]$, the optimal rate is then achievable as long as $A(\boldsymbol{z})\boldsymbol{\theta}^*$ is not concentrated around $\boldsymbol{0}$, and $\mu \to 0$ is properly chosen.

**Assumption 4.4** (Widespread $A(\boldsymbol{z})\boldsymbol{\theta}^*$). Let $\boldsymbol{z} \sim \mathbb{P}$ and let $\|\boldsymbol{x}\|_{-\infty} := \min\{|x_1|, \dots, |x_m|\}$ denote the minimal absolute value among elements of $\boldsymbol{x}$. We assume there exists a threshold $\tau > 0$ such that $\mathbb{P}_{\boldsymbol{z}}\left[\|A(\boldsymbol{z})\boldsymbol{\theta}^*\|_{-\infty} \ge t\right] \ge 1 - \exp\left(-1/t^2\right), \forall t \le \tau.$

Assumption 4.4 assumes that $A(\boldsymbol{z})\boldsymbol{\theta}^*$ is at least $t$-distance away from nondifferentiable points with certain probability; otherwise, if all $A(\boldsymbol{z})\boldsymbol{\theta}^*$ are nondifferentiable points, this distribution of $\boldsymbol{z}$ might be ill-posed and impractical. This assumption is motivated by observations in neural networks with ReLU activation functions where $A\boldsymbol{\theta}^*$ are often far from nondifferentiable points (Ma & Fattahi, 2022).

**Theorem 4.5** (C-OP Performance). *Suppose Assumptions 2.3 and 4.4 hold. When we have either (i) $\delta \lesssim \min\left\{\exp\left(-\max\{\beta_h^2, m^2/\tau^4\}/d\right), n^{-m^2/d}\right\}$, or (ii) $\delta \lesssim \exp\left(-\max\{\beta_h^2, m^2/\tau^4\}/d\right)$ and*

$\varepsilon \gtrsim \sqrt{m^2 \ln(n)/n}$, *then running* C-OP *with Gaussian kernel and parameters in Lemma 3.3 yields*

$$\mathcal{R}(\mathcal{A}; \mathbb{P}) \leq 8\sqrt{2}CR \cdot \left( \frac{1}{\sqrt{n}} + \frac{\sqrt{d \ln(1/\delta)}}{n\varepsilon} \right).$$

The theorem claims that under some assumptions, running C-OP can achieve the same optimal convergence rate as that by Noisy-SGD (Bassily et al., 2019; 2020). However, readers should be reminded that the optimal rate of C-OP comes at prices of (i) some restrictions on $(\varepsilon, \delta)$ and (ii) a smaller set of admissible distributions. Practically, both requirements are mild. Our numerical experiments keep showing satisfactory performance of C-OP, even when these requirements are not necessarily met.

### 4.3 SOME SPECIAL CASES

We found that some special loss functions not strictly following the assumed structure also benefit from convolution. Applying C-OP to those functions can achieve optimal rates if the distribution is not ill-posed (see Appendix B.4 for details).

**Piecewise Linear Loss** (Figure 4, middle). Suppose the nonsmooth loss function is piecewise linear with $P < \infty$ pieces in the form $f(\boldsymbol{\theta}; \boldsymbol{z}) := \max_{p \in [P]} \{\langle \boldsymbol{a}_p, A(\boldsymbol{z})\boldsymbol{\theta} \rangle + b_p\}$ where $\{\boldsymbol{a}_p, b_p\}_{p=1}^P$ are known parameters of pieces. In this case, the smooth approximation is $f_\mu(\boldsymbol{\theta}; \boldsymbol{z}) = \mathbb{E}_{\mathbf{k}} \left[ \max_{p \in [P]} \{\langle \boldsymbol{a}_p, A(\boldsymbol{z})\boldsymbol{\theta} + \mu\mathbf{k} \rangle + b_p\} \right]$.

**Bowl-shaped Loss** (Figure 4, rightmost) Suppose bowl-shaped thresholding loss function $f(\boldsymbol{\theta}; z) := (\|A\boldsymbol{\theta}\|_1 - z)^+$ with known $A \in \mathbb{R}^{m \times d}$. In this case, applying convolution gives $f_\mu(\boldsymbol{\theta}; z) = \mathbb{E}_{\mathbf{k}} \left[ (\|A\boldsymbol{\theta} + \mu\mathbf{k}\|_1 - z)^+ \right]$.

### 4.4 COMPARE TO (GENERALIZED) MOREAU ENVELOPE

Because our work is motivated by DP convex optimization, we are interested in comparing convolution with (generalized) Moreau Envelope (Parikh et al., 2014), which is the most common smoothing approach in DP literature, see its applications in Bassily et al. (2014; 2019), Feldman et al. (2020); Asi et al. (2021); Bassily et al. (2022). (Standard) Moreau Envelope approximates the original nonsmooth function $g : \mathbb{R}^m \to \mathbb{R}$ with a smooth approximation obtained from a minimization problem involving a smooth function $\phi_\mu(\cdot) := \mu\phi(\cdot/\mu)$ with $\phi(\cdot) = \frac{1}{2} \|\cdot\|_2^2$ and $\mu = 1$,

$$g_{\mathsf{ME}}(\boldsymbol{x}) = \inf_{\boldsymbol{u} \in \mathbb{R}^m} \{g(\boldsymbol{u}) + \phi_\mu(\boldsymbol{x} - \boldsymbol{u})\}, \quad \forall \boldsymbol{x}. \tag{6}$$

The Generalized Moreau takes other $\phi(\cdot)$ function and possesses similar properties to convolution.

**Lemma 4.6** (Properties of $g_{\mathsf{ME}}$, partially from Beck & Teboulle (2012)). *Let $g : \mathbb{R}^m \to \mathbb{R}$ be a closed, proper, convex, and $L$-Lipschitz continuous function (w.r.t. $\|\cdot\|_2$), and let $\phi : \mathbb{R}^m \to \mathbb{R}$ be a $\beta'$-smooth function satisfying regularity conditions (Condition B.1 in Appendix). Then the Generalized Moreau Envelope $g_{\mathsf{ME}}$ possesses following properties:*

1. *$g_{\mathsf{ME}}$ is convex, $L$-Lipschitz, and $(\beta'/\mu)$-smooth, w.r.t. $\|\cdot\|_2$;*

2. *$\nabla g_{\mathsf{ME}}(\boldsymbol{x}) = \nabla\phi_\mu(\boldsymbol{x} - \boldsymbol{u}^*(\boldsymbol{x}))$, where $\boldsymbol{u}^*(\boldsymbol{x})$ is the minimizer to the r.h.s problem of Eq.(6);*

3. *Let $\phi^\star : \mathbb{R}^m \to \mathbb{R}$ be the Fenchel conjugate of $\phi$, and let $\|\phi^\star\|_\infty := \sup_{\boldsymbol{y} \in \mathcal{B}(L)} \phi^\star(\boldsymbol{y})$. Then, the approximation error is $-\mu \|\phi^\star\|_\infty \leq g_{\mathsf{ME}}(\boldsymbol{x}) - g(\boldsymbol{x}) \leq \mu\phi(\mathbf{0})$.*

It is self-evident that Lemma 4.6 is an analogue of Lemma 3.1. As a minor contribution, the Lipschitz constant of $g_{\mathsf{ME}}$ is tightened from $2L$ (Bassily et al., 2019) to $L$. Comparing Lemmas 3.1 and 4.6, we observe some distinctions between convolution and Moreau.

First, Moreau approximates the original function from below, contrasting with convolution that approximates from above (Figure 3, first and third plots). Approximating from below invalidates the newly developed risk decomposition in Eq.(2); thus analysis developed in this work does not directly apply to Moreau.

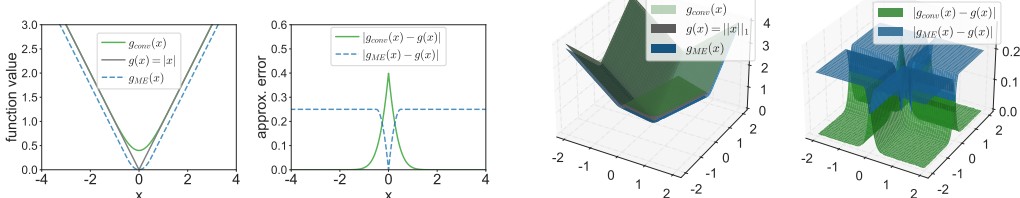

Figure 3: Comparison between Moreau $g_{\mathsf{ME}}$ and convolution $g_{\mathrm{conv}}$. Left two figures: $g(x) = |x|$; right two figures: $g(\boldsymbol{x}) = \|\boldsymbol{x}\|_1$ in a 2-dim space.

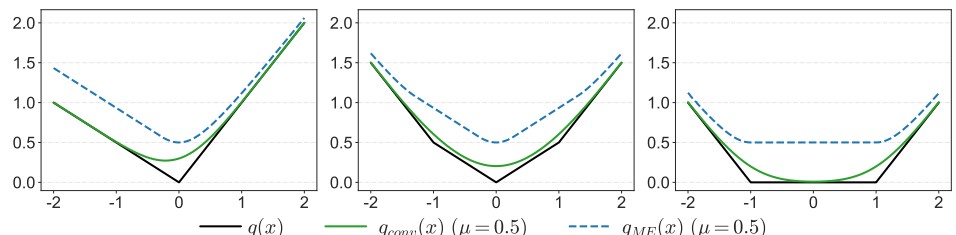

Figure 4: From left to right are quantile, piecewise linear, and bowl-shaped functions (in black), and their smooth approximation functions via (i) convolution with Gaussian kernel (in green) and (ii) Moreau with $\phi(x) = \sqrt{1 + x^2}$ (in blue). Both $g_{\mathsf{ME}}$ and $g_{\mathrm{conv}}$ are 2-smooth.

Second, Moreau approximates poorly at most points, whereas convolution approximates tightly at most points, see Figure 3, second and fourth plots. This suggests that the overall approximation error by Moreau is roughly at the same magnitude as the uniform bound; thus, Moreau cannot enjoy benefits from replacing a uniform bound with pointwise bounds.

If Moreau takes other $\phi$ functions, such as $\phi(\cdot) = L\sqrt{1 + \|\cdot\|_2^2}$, then $g_{\mathsf{ME}}$ can approximate $g$ from above. Nevertheless, the approximation quality is much lower, as shown in Figure 4. Therefore, we prefer convolution over Moreau Envelope. We also highlight that the insights drawn and distinctive features of convolution may have broader impact on other applications.

## 5 EXPERIMENTS

We run experiments on two problems (i) high-dimensional medians $f(\boldsymbol{\theta}; \boldsymbol{y}, A) = \|\boldsymbol{y} - A\boldsymbol{\theta}\|_1$ whose convolution is given in Eq.(1); (ii) piecewise linear $f(\boldsymbol{\theta}; \boldsymbol{y}, A) = \max_{p \in [P]}\{\langle \boldsymbol{a}_p, \boldsymbol{y} - A\boldsymbol{\theta}\rangle + b_p\}$ whose convolution is given in Section 4.3. We use relative risks $\frac{F(\widehat{\boldsymbol{\theta}}^{\mathcal{A}}) - F(\boldsymbol{\theta}^*)}{F(\boldsymbol{\theta}^*)} \times 100\%$ as the performance metric and compare three algorithms; namely, our algorithm C-OP; Moreau Envelope (Bassily et al. 2019, Algorithm 1); and Noisy-SGD (Bassily et al. 2020, Algorithm 2). Noisy-SGD does not have a smoothing step, while the other two have. Figure 5 shows results for problem (i).

It is evident that our algorithm C-OP outperforms existing methods in high-privacy regimes (subplots on the left). In other regimes, it still performs comparably well to Noisy-SGD. Intuitively, the improved performance of C-OP is because of our better utilization of the $\ell_1$ structure, whereas Noisy-SGD is an indiscriminate approach.

## 6 CONCLUSION

**Limitation of our work.** Our work is not without limitations. First, Assumption 4.4 might be hard to verify in practice as it requires the knowledge of $\boldsymbol{\theta}^*$. Second, the assumed model is still restrictive to some extent. For example, terms $A(\boldsymbol{z})\boldsymbol{\theta}$ and $\boldsymbol{z}^\top\boldsymbol{\theta}$ are assumed to be low-dimensional; otherwise, the convergence rate will blow up by an additional factor of $\sqrt{d}$. While this limitation is not unique to ours but is inherent to OP, we would like to bring this issue to the community's attention for further research.

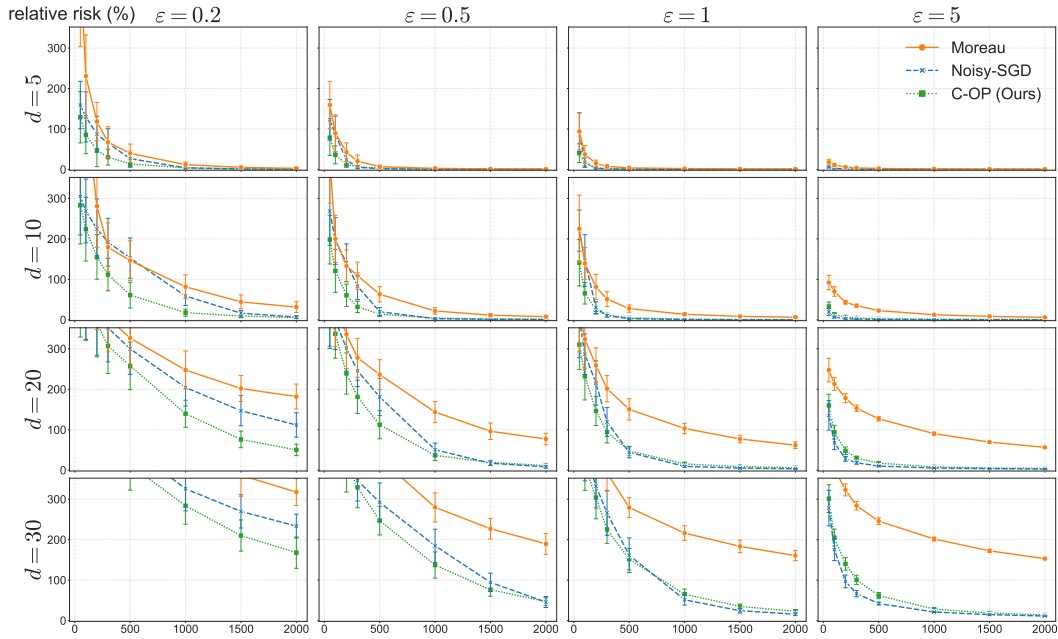

Figure 5: Relative risk v.s. sample size under various settings. Datapoint $\boldsymbol{y} \sim \mathcal{N}(A\boldsymbol{\theta}, \boldsymbol{I}_{3\times 3})$, where for the base case $d = 5$, we let $\boldsymbol{\theta} = (.5, -.5, 1, -1, 1)$, and let each element of matrix $A$ follow $\mathcal{N}(\mu_{A_{ij}}, 1^2)$ with $\mu_A := \begin{bmatrix} 1 & .5 & 0 & 0 & 1 \\ .5 & .5 & 0 & 0 & 1 \\ 0 & 0 & -.5 & 0 & 1 \end{bmatrix}$. For higher dimensional cases, we concat multiply $A$s and $\boldsymbol{\theta}$s. Results are averaged from 50 runs. Error bar = std. More results in Appendix B.7.

In this paper, we studied how to apply OP to nonsmooth DP-SCO problems whose loss function has an implicit $\ell_1$ structure. We proposed to wrap OP with an additional convolution smoothing step. Convolution found many distinctive features that make it more suitable than common methods such as Moreau Envelope. These features facilitate tighter analysis of generalization risks, and thus under mild assumptions, convolution-then-OP can achieve optimal rates. Numerical experiments further showcase competitive performance. There are many interesting directions to explore in the future, such as extending the idea in this work to more general nonsmooth functions, and how to get rid of the mild assumptions on groundtruth distributions.

ACKNOWLEDGMENTS

We sincerely thank the anonymous Area Chair and reviewers for their valuable feedback that significantly improved this work. This research is supported by the Ministry of Education, Singapore, under its MOE AcRF Tier 1 (RG117/23) grant.

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

## A  OMITTED MATERIALS AND PROOFS FOR SECTION 3

### A.1  CONDITIONS ON KERNEL FUNCTIONS

*Condition* A.1 (Kernel Functions). Let $k : \mathbb{R}^m \to \mathbb{R}_+$ be a nonnegative function defined on $d$-dimensional real space. We assume the function $k$ has following properties:

- **Integrate to 1**: $\int_{\mathbb{R}^d} k(\boldsymbol{v}) \, d\boldsymbol{v} = 1$;
- **Central Symmetry**: $k(\boldsymbol{v}) = k(-\boldsymbol{v}), \forall \boldsymbol{v} \in \mathbb{R}^m$;
- **Monotonicity**: $k(\boldsymbol{v})$ is decreasing in $\|\boldsymbol{v}\|_p$ for some $p \geq 1$;
- **Finite Moments**: $\kappa_2 := \int_{\mathbb{R}^m} \|\boldsymbol{v}\| \, k(\boldsymbol{v}) \, d\boldsymbol{v} < \infty, \bar{k} := \sup_{\boldsymbol{v} \in \mathbb{R}^m} k(\boldsymbol{v}) = k(\boldsymbol{0}) < \infty.$

### A.2  PROOF OF LEMMA 3.1: PROPERTIES OF CONVOLUTION

*Proof.*      1. Convexity and Lipschitzness can be easily shown by definition:

$$\text{(convexity)} \quad g_\mu(\lambda \boldsymbol{x} + (1-\lambda)\boldsymbol{y}) \leq \int [\lambda g(\boldsymbol{x} + \mu\boldsymbol{v}) + (1-\lambda)g(\boldsymbol{y} + \mu\boldsymbol{v})] \cdot k(\boldsymbol{v}) \, d\boldsymbol{v}$$
$$= \lambda g_\mu(\boldsymbol{x}) + (1-\lambda)g_\mu(\boldsymbol{y});$$
$$\text{(Lipschitzness)} \quad g_\mu(\boldsymbol{x}) - g_\mu(\boldsymbol{y}) = \int [g(\boldsymbol{x} + \mu\boldsymbol{v}) - g(\boldsymbol{y} + \mu\boldsymbol{v})] \, k(\boldsymbol{v}) \, d\boldsymbol{v} \leq L \|\boldsymbol{x} - \boldsymbol{y}\|_p.$$

To show smoothness, we temporarily assume the second property that $\nabla g_\mu(\boldsymbol{x}) = \mathbb{E}_{\mathbf{k}} [\nabla f(\boldsymbol{x} + \mu\mathbf{k})]$ is true. Let $\mathsf{TV}(\mathbb{P}, \mathbb{Q})$ and $\mathsf{KL}(\mathbb{P}, \mathbb{Q})$ denote the total variation and KL-divergence between distributions $\mathbb{P}$ and $\mathbb{Q}$. Let $q > 0$ be conjugate index such that $1/p + 1/q = 1$. By definition of smoothness, it suffices to show $\|\nabla g_\mu(\boldsymbol{x}) - \nabla g_\mu(\boldsymbol{y})\|_q \leq \beta_\mu \|\boldsymbol{x} - \boldsymbol{y}\|_p$. Direct computation gives

$$\|\nabla g_\mu(\boldsymbol{x}) - \nabla g_\mu(\boldsymbol{y})\|_q = \left\| \int [\nabla g(\boldsymbol{x} + \mu\boldsymbol{v}) - \nabla g(\boldsymbol{y} + \mu\boldsymbol{v})] \, k(\boldsymbol{v}) \, d\boldsymbol{v} \right\|_q$$

$$= \left\| \int \nabla g(\mu\boldsymbol{u}) \cdot [k(\boldsymbol{u} - \boldsymbol{x}/\mu) - k(\boldsymbol{u} - \boldsymbol{y}/\mu) \, d\boldsymbol{u}] \right\|_q \quad \text{(change variables)}$$

$$\leq \cdot \int \|\nabla g(\mu\boldsymbol{u})\|_q \cdot |k(\boldsymbol{u} - \boldsymbol{x}/\mu) - k(\boldsymbol{u} - \boldsymbol{y}/\mu)| \, d\boldsymbol{u} \quad \text{(triangle ineq.)}$$

$$\leq L \cdot \int |k(\boldsymbol{u} - \boldsymbol{x}/\mu) - k(\boldsymbol{u} - \boldsymbol{y}/\mu)| \, d\boldsymbol{u} \quad (g \text{ is } L\text{-lips cts w.r.t. } \|\cdot\|_p)$$

$$= 2L \cdot \frac{1}{2} \int |k(\boldsymbol{u}) - k(\boldsymbol{u} + (\boldsymbol{x} - \boldsymbol{y})/\mu)| \, d\boldsymbol{u}$$

$$= 2L \cdot \mathsf{TV}(\mathbf{k}, \mathbf{k} + \boldsymbol{\delta}/\mu)). \tag{7}$$

$$\leq 2L \cdot \sqrt{\frac{1}{2} \mathsf{KL}(\mathbf{k}, \mathbf{k} + \boldsymbol{\delta}/\mu)} \tag{8}$$

The integral in the third-to-last line is the total variation ($\mathsf{TV}$) between random variables $\mathbf{k}$ and $\mathbf{k} + \boldsymbol{\delta}/\mu$ for any given $\boldsymbol{\delta} := \boldsymbol{y} - \boldsymbol{x}$ and $\mu$; the last line is by Pinsker's inequality. Therefore, it suffices to control the KL-divergence between $\mathbf{k}$ and $\mathbf{k} + \boldsymbol{\delta}/\mu$.

(a) When $\mathbf{k} \sim \mathcal{N}(\boldsymbol{0}, \boldsymbol{I}_{d \times d})$, the KL-divergence between two Gaussians are well known:

$$\mathsf{KL}(\mathcal{N}(\boldsymbol{0}, \boldsymbol{I}), \mathcal{N}(\boldsymbol{\delta}/\mu, \boldsymbol{I})) = \frac{\|\boldsymbol{\delta}\|_2^2}{2\mu^2},$$

see Feldman et al. (2018, Theorem 33). Consequently, $\|\nabla g_\mu(\boldsymbol{x}) - \nabla g_\mu(\boldsymbol{y})\|_q \leq \frac{L}{\mu} \|\boldsymbol{x} - \boldsymbol{y}\|_2 \leq \frac{L}{\mu} \|\boldsymbol{x} - \boldsymbol{y}\|_p, \forall p \in [1, 2].$

(b) When the kernel function is Exponential $k(\boldsymbol{v}) = \frac{1}{\mathfrak{n}} \cdot e^{-\|\boldsymbol{v}\|_2}$ with $\mathfrak{n} = \frac{2\pi^{d/2}\Gamma(d)}{\Gamma(d/2)}$ where $\Gamma(z) = \int_0^\infty t^{z-1} e^{-z} \, dt$ is the gamma function, we consider two cases:

i. when $\|\boldsymbol{\delta}\|_p \geq \mu$: we can show

$$\mathsf{KL}(\mathbf{k}, \mathbf{k} + \boldsymbol{\delta}/\mu) = \int \frac{1}{\mathfrak{n}} e^{-\|\boldsymbol{v}\|_2} \cdot \ln\left(\frac{e^{-\|\boldsymbol{v}\|_2}}{e^{-\|\boldsymbol{v}-\boldsymbol{\delta}/\mu\|_2}}\right) d\boldsymbol{v}$$

$$\leq \int \frac{1}{\mathfrak{n}} e^{-\|\boldsymbol{v}\|_2} \cdot \|\boldsymbol{\delta}\|_2 /\mu \, d\boldsymbol{v}$$

$$= \|\boldsymbol{\delta}\|_2 /\mu \leq \|\boldsymbol{\delta}\|_p /\mu. \tag{9}$$

Therefore $\|\nabla g_\mu(\boldsymbol{x}) - \nabla g_\mu(\boldsymbol{y})\|_q \leq 2L\sqrt{\frac{1}{2}\|\boldsymbol{\delta}\|_p /\mu} \leq \sqrt{2}L\|\boldsymbol{\delta}\|_p /\mu$, where the second inequality is from $\|\boldsymbol{\delta}\|_p /\mu \geq 1$.

ii. when $\|\boldsymbol{\delta}\|_p \leq \mu$: for notational brevity, we temporarily use $\mathbb{P}, \mathbb{Q}$ to denote the probability measure of $\mathbf{k}, \mathbf{k} + \boldsymbol{\delta}/\mu$ respectively. By the inequality between KL-divergence and $\chi^2$-divergence $\mathsf{KL}(\mathbb{P}, \mathbb{Q}) \leq D_{\chi^2}(\mathbb{P}, \mathbb{Q})$ and the fact that $e^x - 1 \leq \sqrt{3}x$ when $x \in [0, 1]$, we can show that

$$\mathsf{KL}(\mathbb{P}, \mathbb{Q}) \leq D_{\chi^2}(\mathbb{P}, \mathbb{Q}) = \int \left(\frac{e^{-\|\boldsymbol{v}\|_2}}{e^{-\|\boldsymbol{v}-\boldsymbol{\delta}/\mu\|_2}} - 1\right)^2 \cdot \mathbb{Q}(\boldsymbol{v}) \, d\boldsymbol{v}$$

$$\leq \int \left(e^{\|\boldsymbol{\delta}\|_2/\mu} - 1\right)^2 \cdot \mathbb{Q}(\boldsymbol{v}) \, d\boldsymbol{v} \tag{10}$$

$$\leq 3\|\boldsymbol{\delta}\|_2^2 /\mu^2 \leq 3\|\boldsymbol{\delta}\|_p^2 /\mu^2, \tag{11}$$

Consequently, $\|\nabla g_\mu(\boldsymbol{x}) - \nabla g_\mu(\boldsymbol{y})\|_q \leq 2L\sqrt{\frac{1}{2} \cdot 3\|\boldsymbol{\delta}\|_p^2 /\mu^2} = \sqrt{6}L\|\boldsymbol{\delta}\|_p /\mu$.

Combining both cases, we conclude that when exponential kernel function is used, the smoothed approximation $g_\mu$ is $\sqrt{6}L/\mu$-smooth w.r.t. $\|\cdot\|_p$.

(c) When we use Laplacian kernel $k(\boldsymbol{v}) = e^{-\|\boldsymbol{v}\|_1}/2^d$, the analysis idea for Exponential kernel can still apply. Specifically, we can consider two cases $\|\boldsymbol{\delta}\|_1 \geq \mu$ and $\|\boldsymbol{\delta}\|_1 \leq \mu$. In the first case, it can be shown that $\mathsf{KL}(\mathbf{k}, \mathbf{k} + \boldsymbol{\delta}/\mu) \leq \|\boldsymbol{\delta}\|_1 /\mu$ which follows from equation 9; in the second case, we can also show $D_{\chi^2}(\mathbb{P}, \mathbb{Q}) \leq 3\|\boldsymbol{\delta}\|_1^2 /\mu^2$ as that in equation 11. Therefore, $\|\nabla g_\mu(\boldsymbol{x}) - \nabla g_\mu(\boldsymbol{y})\|_q \leq 2L\sqrt{\frac{1}{2} \cdot 3\|\boldsymbol{\delta}\|_1^2 /\mu^2} = \sqrt{6}L\|\boldsymbol{\delta}\|_1 /\mu \leq \sqrt{6m}L\|\boldsymbol{\delta}\|_p /\mu, \forall p \in [1, 2]$.

2. Since we assume loss function $g$ is Lipschitz and convex, it implies $g$ is differentiable almost everywhere; thus, $\nabla g(\boldsymbol{x} + \mu\mathbf{k})$ exists with probability 1. As a result of that,

$$\nabla g_\mu(\boldsymbol{x}) = \nabla \int g(\boldsymbol{x} + \mu\boldsymbol{v})k(\boldsymbol{v}) \, d\boldsymbol{v} = \int \nabla g(\boldsymbol{x} + \mu\boldsymbol{v})k(\boldsymbol{v}) \, d\boldsymbol{v} = \mathbb{E}\left[\nabla g(\boldsymbol{x} + \mu\mathbf{k})\right].$$

3. Lower bound: for any $\boldsymbol{x} \in \mathcal{X}$,

$$g_\mu(\boldsymbol{x}) = \mathbb{E}_{\mathbf{k}}\left[g(\boldsymbol{x} + \mu \cdot \mathbf{k})\right] \geq g(\boldsymbol{x} + \mu \cdot \mathbb{E}[\mathbf{k}]) = g(\boldsymbol{x}),$$

where the inequality is by Jensen's inequality, and the last equality is from the fact that $\mathbb{E}[\mathbf{k}] = \mathbf{0}$ since $\mathbf{k}$ is centrally symmetric.

Upper bound: for any $\boldsymbol{x} \in \mathcal{X}$,

$$g_\mu(\boldsymbol{x}) - g(\boldsymbol{x}) = \int \left[g(\boldsymbol{x} + \mu\boldsymbol{v}) - g(\boldsymbol{x})\right] k(\boldsymbol{v}) \, d\boldsymbol{v} \leq L\mu \int \|\boldsymbol{v}\|_p k(\boldsymbol{v}) \, d\boldsymbol{v} =: L\mu\kappa_p,$$

where $\kappa_p := \int \|\boldsymbol{v}\|_p k(\boldsymbol{v}) \, d\boldsymbol{v} = \mathbb{E}_{\mathbf{k}}\left[\|\mathbf{k}\|_p\right]$.

Specially, we do some calculations for $\kappa_2$ as we will use $\kappa_2$ later. Denote the surface area of a given set by $S(\cdot)$. A well-known result from the geometry literature is that the surface area of a $m$-dimensional Euclidean ball with radius $t$ is $S(\mathcal{B}(t)) = \frac{2\pi^{m/2}}{\Gamma(m/2)} t^{m-1}$.

(a) When using Gaussian kernel $\mathbf{k} \sim \mathcal{N}(\mathbf{0}, \boldsymbol{I})$, we have $\kappa_2 = \mathbb{E}\left[\|\mathbf{k}\|_2\right] \leq \sqrt{\mathbb{E}\left[\|\mathbf{k}\|_2^2\right]} = \sqrt{m}$, which is by Jensen's inequality and by noticing that $\|\mathbf{k}\|^2$ is a chi-square random variable with degree $m$.

(b) When using exponential kernel $k(\boldsymbol{v}) = \frac{1}{\mathfrak{n}} \cdot e^{-\|\boldsymbol{v}\|_2}$ with $\mathfrak{n} = \frac{2\pi^{m/2}\Gamma(m)}{\Gamma(m/2)}$, we can show that

$$
\begin{aligned}
\kappa_2 &= \int_{\mathbb{R}^m} \|\boldsymbol{v}\|_2 \cdot \frac{1}{\mathfrak{n}} e^{-\|\boldsymbol{v}\|_2} \, m\boldsymbol{v} = \frac{1}{\mathfrak{n}} \int_0^\infty t e^{-t} S(\mathcal{B}(t)) \, dt \\
&= \frac{1}{\mathfrak{n}} \cdot \int_0^\infty t e^{-t} \frac{2\pi^{m/2}}{\Gamma(m/2)} t^{m-1} \, dt = \frac{\Gamma(m+1)}{\Gamma(m)} = m.
\end{aligned}
$$

(c) When using Laplacian kernel $k(\boldsymbol{v}) = e^{-\|\boldsymbol{v}\|_1}/2^m$,

$$
\begin{aligned}
\kappa_2 &= \int_{\mathbb{R}^m} \|\boldsymbol{v}\|_2 \cdot \frac{e^{-\|\boldsymbol{v}\|_1}}{2^m} \, m\boldsymbol{v} \leq \int_{\mathbb{R}^m} \|\boldsymbol{v}\|_1 \cdot \frac{e^{-\|\boldsymbol{v}\|_1}}{2^m} \, d\boldsymbol{v} \\
&= \sum_{j=1}^m \left( \int_{-\infty}^\infty |v_j| \cdot \frac{e^{-|v_j|}}{2} \, dv_j \right) \quad \text{(integrate layer by layer)} \\
&= m.
\end{aligned}
$$

4. We then notice that, the central symmetry of kernel function $k(\cdot)$ allows another representation of the approximation gap, for any $\boldsymbol{x} \in \mathcal{X}$:

$$
g_\mu(\boldsymbol{x}) = \int_{\boldsymbol{v} \in \mathbb{R}^m} g(\boldsymbol{x} + \mu\boldsymbol{v}) k(\boldsymbol{v}) \, d\boldsymbol{v} \tag{12}
$$

$$
= \int_{-\boldsymbol{v}' \in \mathbb{R}^m} g(\boldsymbol{x} - \mu\boldsymbol{v}') k(-\boldsymbol{v}') \, d(-\boldsymbol{v}'), \quad \text{(change variables } \boldsymbol{v}' := -\boldsymbol{v})
$$

$$
= \int_{\boldsymbol{v}' \in \mathbb{R}^m} g(\boldsymbol{x} - \mu\boldsymbol{v}') k(\boldsymbol{v}') d(\boldsymbol{v}') \quad (k(\cdot) \text{ is central symmetric}) \tag{13}
$$

Combining equation 12 and equation 13 gives

$$
\begin{aligned}
g_\mu(\boldsymbol{x}) &= \frac{1}{2} \left( \int_{\boldsymbol{v} \in \mathbb{R}^m} g(\boldsymbol{x} + \mu\boldsymbol{v}) k(\boldsymbol{v}) \, d\boldsymbol{v} + \int_{\boldsymbol{v} \in \mathbb{R}^m} g(\boldsymbol{x} - \mu\boldsymbol{v}) k(\boldsymbol{v}) \, d\boldsymbol{v} \right) \\
&= \int_{\boldsymbol{v} \in \mathbb{R}^m} \left[ \frac{g(\boldsymbol{x} + \mu\boldsymbol{v}) + g(\boldsymbol{x} - \mu\boldsymbol{v})}{2} \right] k(\boldsymbol{v}) \, d\boldsymbol{v}
\end{aligned}
$$

$\square$

### A.3 PROOF OF LEMMA 4.1: SMALLER DOMAIN OF INTEGRATION

*Proof.* To prove the statement for the $m$-dimensional case, it suffices to prove the case of 1-dimension, i.e., to prove $\mathcal{V}_\mu(x) := \{v \in \mathbb{R} : \frac{|x+\mu v|+|x-\mu v|}{2} - |x| > 0\} = \{v : \mathbb{R} : |v| > |x|/\mu\} =: \mathcal{V}_\mu^{\ell_1}(x)$. For ease of notation, we omit the dependence on $x$. We start with the l.h.s:

$$
\begin{aligned}
\mathcal{V}_\mu &= \{v \in \mathbb{R} : |x + \mu v| + |x - \mu v| > 2|x|\} \\
&= \underbrace{\left\{ v \in \mathbb{R} : \begin{matrix} |x+\mu v| + |x-\mu v| > 2|x|; \\ \mu v > |x|. \end{matrix} \right\}}_{=:E_1} \cup \underbrace{\left\{ v \in \mathbb{R} : \begin{matrix} |x+\mu v| + |x-\mu v| > 2|x|; \\ \mu v < -|x|. \end{matrix} \right\}}_{=:E_2} \\
&\cup \underbrace{\left\{ v \in \mathbb{R} : \begin{matrix} |x+\mu v| + |x-\mu v| > 2|x|; \\ 0 \leq \mu v \leq |x|. \end{matrix} \right\}}_{=:E_3} \cup \underbrace{\left\{ v \in \mathbb{R} : \begin{matrix} |x+\mu v| + |x-\mu v| > 2|x| \\ |x| \leq \mu v \leq 0. \end{matrix} \right\}}_{=:E_4}
\end{aligned}
$$

Since $\mathcal{V}_\mu$ is divided into four sets, we can check out each set individually.

- $E_1$: For any $v$ such that $\mu v > |x|$, we have $|x + \mu v| = x + \mu v$ and $|x - \mu v| = \mu v - x$; thus, $|x + \mu v| + |x - \mu v| = 2\mu v > 2|x|$. In other words, $\mu v > |x|$ is sufficient to characterize the set $E_1$, and the another constraint is redundant. So $E_1 = \{v \in \mathbb{R} : \mu v > |x|\}$.

- $E_2$: For any $v$ such that $\mu v < -|x|$, we have $|x + \mu v| = -x - \mu v$ and $|x - \mu v| = x - \mu v$; thus, $|x + \mu v| + |x - \mu v| = -2\mu v > 2|x|$. Similarly, $\mu v < -|x|$ is sufficient. So $E_1 = \{v \in \mathbb{R} : \mu v < -|x|\}$.

- $E_3$: For any $v$ such that $0 \le \mu v \le |x|$, (i) if $x \ge 0$, then $|x + \mu v| + |x - \mu v| = 2x = 2|x|$; (ii) if $x < 0$, then $|x + \mu v| + |x - \mu v| = -2x = 2|x|$. The preceding two cases indicate that no matter what $x$ we have, $|x + \mu v| + |x - \mu v|$ is always strictly equal to $2|x|$. Therefore, $E_3 = \emptyset$ is an empty set.

- $E_4$: Following the same idea for $E_3$, it is easy to show $E_4$ is also an empty set.

Combining four cases, we conclude $\mathcal{V}_\mu = E_1 \cup E_2 = \{v \in \mathbb{R} : |v| > |x|/\mu\} = \mathcal{V}_\mu^{\ell_1}$.

$\square$

## A.4 PROOF OF THEOREM 3.2: PRIVACY GUARANTEE

*Proof.* Let $\mathcal{A}(\mathcal{D}) := \arg\min_{\boldsymbol{\theta}} \frac{1}{n}\sum_{i=1}^n f_\mu(\boldsymbol{\theta}; \boldsymbol{z}_i) + \lambda \|\boldsymbol{\theta}\|_2^2 + \frac{\boldsymbol{b}^\top \boldsymbol{\theta}}{n}$ be the output of $\mathcal{A}_{\text{C-OP}}$. We explicitly indicate its dependence on dataset $\mathcal{D}$. We are going to show, for any $\boldsymbol{v}$ and a pair of neighboring datasets $\mathcal{D} \sim \mathcal{D}'$,

$$\frac{\Pr_{\mathcal{A}}[\mathcal{A}(\mathcal{D}) = \boldsymbol{v}]}{\Pr_{\mathcal{A}}[\mathcal{A}(\mathcal{D}') = \boldsymbol{v}]} \le e^\varepsilon, \quad \text{w.p. at least } 1 - \delta.$$

By first-order-condition, $\boldsymbol{b}(\mathcal{A}(\mathcal{D}); \mathcal{D}) = -\sum_{i=1}^n f_\mu(\mathcal{A}(\mathcal{D}); \boldsymbol{z}_i) - 2n\lambda\mathcal{A}(\mathcal{D})$. Changing variables according to function inverse theorem, the output $\mathcal{A}(\mathcal{D})$ can be represented as a function of $\boldsymbol{b}$ in a probabilistic way; that is $\Pr_{\mathcal{A}}[\mathcal{A}(\mathcal{D}) = \boldsymbol{v}] = pdf(\boldsymbol{b}(\boldsymbol{v}; \mathcal{D})) \cdot |\det(\nabla \boldsymbol{b}(\boldsymbol{v}; \mathcal{D}))|$ for any possible output $\boldsymbol{v}$. Here, on the right-hand-side, $pdf(\boldsymbol{b}(\cdot; \mathcal{D}))$ is the pdf of noise $\boldsymbol{b}$, and $\nabla \boldsymbol{b}$ is a function of $\boldsymbol{v}$; $\det(\cdot)$ is the determinant of a given matrix. Therefore, we must have

$$\frac{\Pr_{\mathcal{A}}[\mathcal{A}(\mathcal{D}) = \boldsymbol{v}]}{\Pr_{\mathcal{A}}[\mathcal{A}(\mathcal{D}') = \boldsymbol{v}]} = \frac{pdf(\boldsymbol{b}(\mathcal{A}(\mathcal{D}); \mathcal{D}))}{pdf(\boldsymbol{b}(\mathcal{A}(\mathcal{D}); \mathcal{D}'))} \cdot \frac{|\det(\nabla \boldsymbol{b}(\mathcal{A}(\mathcal{D}); \mathcal{D}))|}{|\det(\nabla \boldsymbol{b}(\mathcal{A}(\mathcal{D}); \mathcal{D}'))|}, \quad \forall \boldsymbol{v}, \tag{14}$$

Without loss of generality, we assume $\mathcal{D}'$ has one more entry $\boldsymbol{z}'$ than $\mathcal{D}$, which immediately implies

$$\boldsymbol{b}(\mathcal{A}(\mathcal{D}); \mathcal{D}') = \boldsymbol{b}(\mathcal{A}(\mathcal{D}); \mathcal{D}) + \nabla f_\mu(\mathcal{A}(\mathcal{D}); \boldsymbol{z}_n).$$

Recall the smoothed function is

$$f_\mu(\boldsymbol{\theta}; \boldsymbol{z}) = \mathbb{E}_{\mathbf{k}}[\|A(\boldsymbol{z})\boldsymbol{\theta} + \mu\mathbf{k}\|_1] + h(\boldsymbol{z}^\top \boldsymbol{\theta}).$$

Its gradient at any given $\boldsymbol{\theta}$ is, by part 2 of Lemma 3.1,

$$\begin{aligned}
\nabla f_\mu(\boldsymbol{\theta}; \boldsymbol{z}) &= \mathbb{E}_{\mathbf{k}}\left[\nabla_{\boldsymbol{\theta}}\left(\|A(\boldsymbol{z})\boldsymbol{\theta} + \mu\mathbf{k}\|_1 + h(\boldsymbol{z}^\top \boldsymbol{\theta})\right)\right] \\
&= \mathbb{E}_{\mathbf{k}}\left[A(\boldsymbol{z})^\top \text{sgn}(A(\boldsymbol{z})\boldsymbol{\theta} + \mu\mathbf{k}) + \boldsymbol{z}h'(\boldsymbol{z}^\top \boldsymbol{\theta})\right] \\
&= A(\boldsymbol{z})^\top \mathbb{E}_{\mathbf{k}}[\text{sgn}(A(\boldsymbol{z})\boldsymbol{\theta} + \mu\mathbf{k})] + \boldsymbol{z}h'(\boldsymbol{z}^\top \boldsymbol{\theta}),
\end{aligned}$$

where $\text{sgn}(\cdot)$ is the sign vector.

Remember that, the noise $\boldsymbol{b}(\mathcal{A}(\mathcal{D}); \mathcal{D}) \sim \mathcal{N}(\boldsymbol{0}, \sigma^2 \boldsymbol{I})$. Thus, $\boldsymbol{b}(\mathcal{A}(\mathcal{D}); \mathcal{D}') \sim \mathcal{N}(\nabla f_\mu(\mathcal{A}(\mathcal{D}); \boldsymbol{z}_n), \sigma^2 \boldsymbol{I})$. Their likelihood ratio thus becomes

$$\begin{aligned}
\frac{pdf(\boldsymbol{b}(\mathcal{A}(\mathcal{D}); \mathcal{D}))}{pdf(\boldsymbol{b}(\mathcal{A}(\mathcal{D}); \mathcal{D}'))} &= \frac{\exp\left(-\frac{1}{2}\|\boldsymbol{b}(\mathcal{A}(\mathcal{D}); \mathcal{D})\|_2^2/\sigma^2\right)}{\exp\left(-\frac{1}{2}\|\boldsymbol{b}(\mathcal{A}(\mathcal{D}); \mathcal{D}) - \nabla f_\mu(\mathcal{A}(\mathcal{D}); \boldsymbol{z}_n)\|_2^2/\sigma^2\right)} \\
&= \exp\left(\left[-\langle \boldsymbol{b}(\mathcal{A}(\mathcal{D}); \mathcal{D}), \nabla f_\mu(\mathcal{A}(\mathcal{D}); \boldsymbol{z}_n)\rangle + \frac{1}{2}\|\nabla f_\mu(\mathcal{A}(\mathcal{D}); \boldsymbol{z}_n)\|_2^2\right]/\sigma^2\right).
\end{aligned} \tag{15}$$

It should be noticed that $\boldsymbol{b}(\mathcal{A}(\mathcal{D}); \mathcal{D})$ and $\nabla f_\mu(\mathcal{A}(\mathcal{D}); \boldsymbol{z}_n)$ are not independent, whereas Kifer et al. (2012) claims they are independent, which is incorrect. This has been fixed by Redberg et al. (2023) for linear models but not for models we considered here. There is a necessity to do the proof ourselves.

We first look at the inner product in Eq.(15):

$$\langle \boldsymbol{b}(\mathcal{A}(\mathcal{D}); \mathcal{D}), \nabla f_\mu(\mathcal{A}(\mathcal{D}); \boldsymbol{z}_n) \rangle = \langle \boldsymbol{b}(\mathcal{A}(\mathcal{D}); \mathcal{D}), A(\boldsymbol{z}_n)^\top \mathbb{E}_{\mathbf{k}} [\mathrm{sgn}(A(\boldsymbol{z}_n)\mathcal{A}(\mathcal{D}) + \mu\mathbf{k})] + \boldsymbol{z}_n h'(\boldsymbol{z}_n^\top \mathcal{A}(\mathcal{D})) \rangle.$$

Because $\boldsymbol{b}(\mathcal{A}(\mathcal{D}; \mathcal{D})) \sim \mathcal{N}(\boldsymbol{0}, \sigma^2 \boldsymbol{I})$, we know $A(\boldsymbol{z}_n)\boldsymbol{b}(\mathcal{A}(\mathcal{D}; \mathcal{D})) \sim \mathcal{N}(\boldsymbol{0}, A(\boldsymbol{z}_n)^\top A(\boldsymbol{z}_n)\sigma^2)$. Moreover, since we assume the 2-norm of $A(\boldsymbol{z})$ is uniformly upper bounded by $\overline{A}$ for all $\boldsymbol{z}$, by the fact that $|\mathrm{sgn}(\cdot)| \le 1$, we have

$$\mathsf{Var}\left[\langle \boldsymbol{b}(\mathcal{A}(\mathcal{D}); \mathcal{D}), A(\boldsymbol{z}_n)^\top \mathbb{E}_{\mathbf{k}}[\mathrm{sgn}(A(\boldsymbol{z}_n)\mathcal{A}(\mathcal{D}) + \mu\mathbf{k})]\rangle\right] \le \mathsf{Var}\left[\langle \mathcal{N}(\boldsymbol{0}, A(\boldsymbol{z})^\top A(\boldsymbol{z})\sigma^2), \mathrm{sgn}(\cdot)\rangle\right]$$
$$\le m\overline{A}^2\sigma^2.$$

Immediately, we know if we use this upper bound, the resulting random variable will have a heavier tail, and the variance can be upper bounded:

$$\mathsf{Var}\left[\langle \mathcal{N}(\boldsymbol{0}, \sigma^2 \boldsymbol{I}), A(\boldsymbol{z}_n)^\top \mathrm{sgn}(\cdot) + \boldsymbol{z}_n h'(\cdot)\rangle\right] \le m\overline{A}^2\sigma^2 + L_h^2\sigma^2\|\boldsymbol{z}_n\|_2^2$$
$$\le m\overline{A}^2\sigma^2 + L_h^2\sigma^2 D^2. \tag{16}$$

Moreover, the bounded 2-norm of matrix $A(\boldsymbol{z})$ also indicates a bounded $\ell_2$-norm of $\nabla f_\mu(\mathcal{A}(\mathcal{D}); \boldsymbol{z})$:

$$\|\nabla f_\mu(\mathcal{A}(\mathcal{D}); \boldsymbol{z})\|_2^2 = \left\|\mathbb{E}_{\mathbf{k}}\left[A(\boldsymbol{z})^\top \mathrm{sgn}(A(\boldsymbol{z})\boldsymbol{\theta} + \mu\mathbf{k}) + \boldsymbol{z}h'(\boldsymbol{z}^\top\boldsymbol{\theta})\right]\right\|_2^2$$
$$\le \mathbb{E}_{\mathbf{k}}\left[\left\|A(\boldsymbol{z})^\top \mathrm{sgn}(A(\boldsymbol{z})\boldsymbol{\theta} + \mu\mathbf{k}) + \boldsymbol{z}h'(\boldsymbol{z}^\top\boldsymbol{\theta})\right\|_2^2\right]$$
$$\le \mathbb{E}_{\mathbf{k}}\left[2\left\|A(\boldsymbol{z})^\top \mathrm{sgn}(A(\boldsymbol{z})\boldsymbol{\theta} + \mu\mathbf{k})\right\|_2^2 + 2\left\|\boldsymbol{z}h'(\boldsymbol{z}^\top\boldsymbol{\theta})\right\|_2^2\right]$$
$$\le 2m\overline{A}^2 + 2D^2 L_h^2. \tag{17}$$

Take log-transformation for both sides of Eq.(15), and then plug (16) and (17) back into (15), we know that the privacy loss random variable $\ln\left(\frac{pdf(\boldsymbol{b}(\mathcal{A}(\mathcal{D}); \mathcal{D}))}{pdf(\boldsymbol{b}(\mathcal{A}(\mathcal{D}); \mathcal{D}'))}\right)$ has a lighter tail than the Gaussian random variable given below

$$\left[\mathcal{N}(0, \sigma^2 \cdot (m\overline{A}^2 + L_h^2 D^2)) + (m\overline{A}^2 + L_h^2 D^2)\right]/\sigma^2.$$

It remains to find a $\sigma^2$ so that $\left[\mathcal{N}(0, \sigma^2 \cdot (m\overline{A}^2 + L_h^2 D^2)) + (m\overline{A}^2 + L_h^2 D^2)\right]/\sigma^2 \le \frac{\varepsilon}{2}$ with probability at least $1 - \delta$. By Gaussian random variable's tail bound $\Pr\left[\mathcal{N}(0, 1^2) \ge \sqrt{2\ln(1/\delta)}\right] \le \delta$, it suffices to set

$$\sigma^2 \ge \frac{C^2 \cdot (8\ln(1/\delta) + 8\varepsilon)}{\varepsilon^2},$$

where $C := \sqrt{m\overline{A}^2 + L_h^2 D^2}$. Therefore, with this $\sigma^2$, we ensure

$$\frac{pdf(\boldsymbol{b}(\boldsymbol{v}; \mathcal{D}))}{pdf(\boldsymbol{b}(\boldsymbol{v}; \mathcal{D}'))} \le e^{\frac{\varepsilon}{2}}, \quad \text{with prob. at least } 1 - \delta. \tag{18}$$

We then come to control the ratio between two determinants in Eq.(14). Denote matrix $E(\boldsymbol{v}) := \nabla \boldsymbol{b}(\boldsymbol{v}; \mathcal{D}) - \nabla \boldsymbol{b}(\boldsymbol{v}; \mathcal{D}') = \nabla^2 f_\mu(\boldsymbol{v}; \boldsymbol{z}_n)$. The rank of matrix $E(\boldsymbol{v})$ over all $\boldsymbol{v} = \mathcal{A}(\mathcal{D})$ is at most $m + 1$. This is because $\nabla^2 f_\mu = A^\top A \cdot \nabla_{\boldsymbol{\theta}} \mathbb{E}_{\boldsymbol{z}}[\mathrm{sgn}(A\boldsymbol{\theta} + \mu\mathbf{k})] + \boldsymbol{z}\boldsymbol{z}^\top h''(\cdot)$. Because $A$ is an $m$-by-$d$ matrix and $m \le d$, the product matrix $A^\top A$ is at most rank-$m$. Further because $\boldsymbol{z}\boldsymbol{z}^\top h''(\cdot)$ is a rank-1 matrix, we conclude that $E$ is at most rank-$(m + 1)$. An immediate result is that the number of different eigenvalues between $\{\rho_i'\}_{i=1}^d$ of matrix $n\nabla^2 \widehat{F}_\mu(\mathcal{A}(\mathcal{D}); \mathcal{D}) + E$ and $\{\rho_i\}_{i=1}^d$ of matrix $n\nabla^2 \widehat{F}_\mu(\mathcal{A}(\mathcal{D}); \mathcal{D})$ is at most $m+1$. Therefore, the ratio between the two determinants below

depends only on the different eigenvalues:

$$
\frac{|\det(\nabla \boldsymbol{b}(\mathcal{A}(\mathcal{D}); \mathcal{D}'))|}{|\det(\nabla \boldsymbol{b}(\mathcal{A}(\mathcal{D}); \mathcal{D}))|} = \frac{\left|\det(-n\nabla^2 \widehat{F}_\mu(\mathcal{A}(\mathcal{D}); \mathcal{D}) - 2n\lambda \boldsymbol{I} - E)\right|}{\left|\det(-n\nabla^2 \widehat{F}_\mu(\mathcal{A}(\mathcal{D}); \mathcal{D}) - 2n\lambda \boldsymbol{I})\right|} = \frac{\Pi_{i=1}^{m+1} |\rho_i' + 2n\lambda|}{\Pi_{i=1}^{m+1} |\rho_i + 2n\lambda|}
$$

$$
\leq \Pi_{i=1}^{m+1} \left(1 + \frac{|\rho_i' - \rho_i|}{2n\lambda}\right)
$$

$$
\leq \left(1 + \frac{\beta_\mu + \beta_h}{2n\lambda}\right)^{m+1}. \tag{19}
$$

The last inequality is due to $(\beta_\mu + \beta_h)$-smoothness of $f_\mu$, which gives $|\rho_i' - \rho_i| \leq \beta_\mu + \beta_h$. A sufficient condition for $(19) \leq e^{\varepsilon/2}$ is $\lambda \geq (\beta_\mu + \beta_h)(m+1)/(n\varepsilon)$. Hence, if $\lambda \geq \frac{(\beta_\mu+\beta_h)(m+1)}{n\varepsilon}$, then

$$
\frac{|\det(\nabla \boldsymbol{b}(\mathcal{A}(\mathcal{D}); \mathcal{D}'))|}{|\det(\nabla \boldsymbol{b}(\mathcal{A}(\mathcal{D}); \mathcal{D}))|} \leq e^{\frac{\varepsilon}{2}}. \tag{20}
$$

Plugging Eqs.(18) and (20) into equation 14, we finally obtain,

$$
\frac{\mathrm{Pr}_\mathcal{A}\left[\mathcal{A}(\mathcal{D}) = \boldsymbol{v}\right]}{\mathrm{Pr}_\mathcal{A}\left[\mathcal{A}(\mathcal{D}') = \boldsymbol{v}\right]} \leq e^\varepsilon, \quad \text{with prob. at least } 1 - \delta,
$$

if $\lambda \geq \frac{(\beta_\mu+\beta_h)(m+1)}{n\varepsilon}$ and $\sigma^2 \geq C^2 \cdot (8\ln(1/\delta) + 8\varepsilon)/\varepsilon^2$ with $C := \sqrt{m\overline{A}^2 + L_h^2 D^2}$. The lowered dependence of $\lambda$ on rank of matrix $\nabla^2 f_\mu$, which is $m+1$ instead of $d$, has also been noticed by Iyengar et al. (2019) and been utilized to improve OP's practicability.

$\square$

**Lemma A.1** (Uniform Stability Lemma, Bousquet & Elisseeff 2002). *Let $\mathcal{A} : \mathcal{Z}^n \to \Theta$ be a $\tau$-uniformly stable algorithm w.r.t. loss function $f : \Theta \times \mathcal{Z} \to \mathbb{R}$. Let $\mathbb{P}$ be a distribution over $\mathcal{Z}$, and $\mathcal{D} \sim \mathbb{P}^n$ be samples i.i.d. drawn from $\mathbb{P}$. Then, we have $\mathbb{E}_{\mathcal{D},\mathcal{A}}\left[F(\mathcal{A}(\mathcal{D})) - \widehat{F}(\mathcal{A}(\mathcal{D}))\right] \leq \tau$.*

### A.5 PROOF OF THEOREM 3.3: PRELIMINARY PERFORMANCE GUARANTEE

*Proof.* Denote $\widehat{F}_\mu^\mathcal{A}(\boldsymbol{\theta}) := \widehat{F}_\mu(\boldsymbol{\theta}) + \lambda \|\boldsymbol{\theta}\|^2 + \frac{\langle \boldsymbol{b}, \boldsymbol{\theta}\rangle}{n}$, and let $\widehat{\boldsymbol{\theta}}^\mathcal{A} := \arg\min_{\boldsymbol{\theta}} \widehat{F}_\mu^\mathcal{A}(\boldsymbol{\theta})$. We first decompose the excess generalization risk of $\mathcal{A}$, i.e., $\mathcal{R}(\mathcal{A}; \mathbb{P}) = \mathbb{E}_{\mathcal{D},\mathcal{A}}\left[F(\boldsymbol{\theta}^\mathcal{A})\right] - F(\boldsymbol{\theta}^*)$, into three parts:

$$
\mathcal{R}(\mathcal{A}; \mathbb{P}) = \mathbb{E}_{\mathcal{D},\mathcal{A}}\left[F(\widehat{\boldsymbol{\theta}}^\mathcal{A}) - \widehat{F}(\widehat{\boldsymbol{\theta}}^\mathcal{A})\right] + \mathbb{E}_{\mathcal{D},\mathcal{A}}\left[\widehat{F}(\widehat{\boldsymbol{\theta}}^\mathcal{A}) - F(\boldsymbol{\theta}^*)\right]
$$

$$
\leq \mathbb{E}_{\mathcal{D},\mathcal{A}}\left[F(\widehat{\boldsymbol{\theta}}^\mathcal{A}) - \widehat{F}(\widehat{\boldsymbol{\theta}}^\mathcal{A})\right] + \mathbb{E}_{\mathcal{D},\mathcal{A}}\left[\widehat{F}_\mu(\widehat{\boldsymbol{\theta}}^\mathcal{A}) - F(\boldsymbol{\theta}^*)\right], \qquad (\text{since } f_\mu \geq f)
$$

$$
= \mathbb{E}_{\mathcal{D},\mathcal{A}}\left[F(\widehat{\boldsymbol{\theta}}^\mathcal{A}) - \widehat{F}(\widehat{\boldsymbol{\theta}}^\mathcal{A})\right] + \mathbb{E}_{\mathcal{D},\mathcal{A}}\left[\widehat{F}_\mu(\widehat{\boldsymbol{\theta}}^\mathcal{A}) - \widehat{F}_\mu(\boldsymbol{\theta}^*)\right] + \mathbb{E}_\mathcal{D}\left[\widehat{F}_\mu(\boldsymbol{\theta}^*) - F(\boldsymbol{\theta}^*)\right]
$$

$$
= \mathbb{E}_{\mathcal{D},\mathcal{A}}\left[F(\widehat{\boldsymbol{\theta}}^\mathcal{A}) - \widehat{F}(\widehat{\boldsymbol{\theta}}^\mathcal{A})\right] + \mathbb{E}_{\mathcal{D},\mathcal{A}}\left[\widehat{F}_\mu(\widehat{\boldsymbol{\theta}}^\mathcal{A}) - \widehat{F}_\mu(\boldsymbol{\theta}^*)\right] + [F_\mu(\boldsymbol{\theta}^*) - F(\boldsymbol{\theta}^*)]. \tag{21}
$$

As a result of the decomposition, it suffices to control three parts separately. The first part can be controlled through uniform stability analysis; the second part can be upper bounded by classic analysis on empirical loss; the last part amounts to approximation error.

1. The first part can be bounded by uniform stability analysis. Specifically, we first notice $f_\mu$ is $L_f := (\sqrt{m}\overline{A} + DL_h)$-Lipschitz continuous w.r.t. $\|\cdot\|_2$. This is because

$$
\partial_\theta f_\mu = A(\boldsymbol{z})^\top \mathrm{sgn}(\cdot) + \boldsymbol{z}h'(\cdot) \implies \|\partial_\theta f_\mu\|_2 \leq \|A(\boldsymbol{z})\|_2 \|\mathrm{sgn}(\cdot)\|_2 + \|\boldsymbol{z}\|_2 L_h
$$

$$
\leq \sqrt{m}\overline{A} + DL_h.
$$

Then, for any given $\mu$, by the facts that $\widehat{F}_\mu^{\mathcal{A}}$ is $2\lambda$-strong convexity and that $f_\mu$ is $L_f :=$ $(m\overline{A} + DL_h)$-Lipschitz continuous, we have:

$$
\begin{aligned}
\lambda \left\| \widehat{\boldsymbol{\theta}}^{\mathcal{A}}(\mathcal{D}) - \widehat{\boldsymbol{\theta}}^{\mathcal{A}}(\mathcal{D}') \right\|_2^2 &\leq \widehat{F}_\mu^{\mathcal{A}}(\widehat{\boldsymbol{\theta}}^{\mathcal{A}}(\mathcal{D}'); \mathcal{D}) - \widehat{F}_\mu^{\mathcal{A}}(\widehat{\boldsymbol{\theta}}^{\mathcal{A}}(\mathcal{D}), \mathcal{D}) \\
&= \widehat{F}_\mu^{\mathcal{A}}(\widehat{\boldsymbol{\theta}}^{\mathcal{A}}(\mathcal{D}'); \mathcal{D}') - \widehat{F}_\mu^{\mathcal{A}}(\widehat{\boldsymbol{\theta}}^{\mathcal{A}}(\mathcal{D}); \mathcal{D}') \\
&\quad + \frac{f_\mu(\widehat{\boldsymbol{\theta}}^{\mathcal{A}}(\mathcal{D}'); \boldsymbol{z}) - f_\mu(\widehat{\boldsymbol{\theta}}^{\mathcal{A}}(\mathcal{D}); \boldsymbol{z})}{n} + \frac{f_\mu(\widehat{\boldsymbol{\theta}}^{\mathcal{A}}(\mathcal{D}); \boldsymbol{z}') - f_\mu(\widehat{\boldsymbol{\theta}}^{\mathcal{A}}(\mathcal{D}'); \boldsymbol{z}')}{n} \\
&\leq \frac{2L_f \cdot \left\| \widehat{\boldsymbol{\theta}}^{\mathcal{A}}(\mathcal{D}) - \widehat{\boldsymbol{\theta}}^{\mathcal{A}}(\mathcal{D}') \right\|_2}{n},
\end{aligned}
$$

which implies $\left\| \widehat{\boldsymbol{\theta}}^{\mathcal{A}}(\mathcal{D}) - \widehat{\boldsymbol{\theta}}^{\mathcal{A}}(\mathcal{D}') \right\|_2 \leq \frac{2L_f}{\lambda n}$. Since function $f$ is also $L_f$-Lipschitz, we can conclude that Algorithm $\mathcal{A}$ is $\frac{2L_f^2}{\lambda n}$ w.r.t. $f$, i.e.,

$$
\left| f(\widehat{\boldsymbol{\theta}}^{\mathcal{A}}(\mathcal{D})) - f(\widehat{\boldsymbol{\theta}}^{\mathcal{A}}(\mathcal{D}')) \right| \leq \frac{2L_f^2}{\lambda n}, \quad \forall \mathcal{D} \sim \mathcal{D}', \forall \boldsymbol{b}, \forall \mu.
$$

Then, by uniform stability lemma A.1, we can conclude that

$$
\mathbb{E}_{\mathcal{D}, \mathcal{A}} \left[ F(\widehat{\boldsymbol{\theta}}^{\mathcal{A}}(\mathcal{D})) - \widehat{F}(\widehat{\boldsymbol{\theta}}^{\mathcal{A}}(\mathcal{D})) \right] \leq \frac{2L_f^2}{\lambda n} \leq \frac{4C^2}{\lambda n}, \tag{22}
$$

where $C := \sqrt{m\overline{A}^2 + D^2 L_h^2}$, a same value as defined in Theorem 3.2.

2. The second part can be upper bounded by empirical loss analysis (Kifer et al., 2012; Iyengar et al., 2019). Let $\widehat{F}_\mu^\#(\boldsymbol{\theta}) := \widehat{F}_\mu(\boldsymbol{\theta}) + \lambda \|\boldsymbol{\theta}\|_2^2$ and let $\widehat{\boldsymbol{\theta}}^\#$ be its minimizer; Firstly, we notice that, by the strong convexity of $\widehat{F}_\mu^{\mathcal{A}}$,

$$
\begin{aligned}
\lambda \left\| \widehat{\boldsymbol{\theta}}^\# - \widehat{\boldsymbol{\theta}}^{\mathcal{A}} \right\|_2^2 &\leq \widehat{F}_\mu^{\mathcal{A}}(\widehat{\boldsymbol{\theta}}^\#) - \widehat{F}_\mu^{\mathcal{A}}(\widehat{\boldsymbol{\theta}}^{\mathcal{A}}) = \widehat{F}_\mu^\#(\widehat{\boldsymbol{\theta}}^\#) - \widehat{F}_\mu^\#(\widehat{\boldsymbol{\theta}}^{\mathcal{A}}) + \frac{\left\langle \boldsymbol{b}, \widehat{\boldsymbol{\theta}}^\# \right\rangle}{n} - \frac{\left\langle \boldsymbol{b}, \widehat{\boldsymbol{\theta}}^{\mathcal{A}} \right\rangle}{n} \\
&\leq \frac{\|\boldsymbol{b}\|_2 \left\| \widehat{\boldsymbol{\theta}}^\# - \widehat{\boldsymbol{\theta}}^{\mathcal{A}} \right\|_2}{n},
\end{aligned}
$$

which implies $\left\| \widehat{\boldsymbol{\theta}}^\# - \widehat{\boldsymbol{\theta}}^{\mathcal{A}} \right\|_2 \leq \frac{\|\boldsymbol{b}\|}{\lambda n}$. Consequently, we can show

$$
\begin{aligned}
\widehat{F}_\mu(\widehat{\boldsymbol{\theta}}^{\mathcal{A}}) - \widehat{F}_\mu(\boldsymbol{\theta}^*) &\leq \left( \widehat{F}_\mu^\#(\widehat{\boldsymbol{\theta}}^{\mathcal{A}}) - \lambda \left\| \widehat{\boldsymbol{\theta}}^{\mathcal{A}} \right\|_2^2 \right) - \left( \widehat{F}_\mu^\#(\widehat{\boldsymbol{\theta}}^\#) - \lambda \|\boldsymbol{\theta}^*\|_2^2 \right) \\
&\leq \left( \widehat{F}_\mu^{\mathcal{A}}(\widehat{\boldsymbol{\theta}}^{\mathcal{A}}) - \frac{\left\langle \boldsymbol{b}, \widehat{\boldsymbol{\theta}}^{\mathcal{A}} \right\rangle}{n} \right) - \left( \widehat{F}_\mu^{\mathcal{A}}(\widehat{\boldsymbol{\theta}}^\#) - \frac{\left\langle \boldsymbol{b}, \widehat{\boldsymbol{\theta}}^\# \right\rangle}{n} \right) + \lambda \|\boldsymbol{\theta}^*\|_2^2 \\
&\leq \frac{\|\boldsymbol{b}\|_2 \left\| \widehat{\boldsymbol{\theta}}^\# - \widehat{\boldsymbol{\theta}}^{\mathcal{A}} \right\|_2}{n} + \lambda \|\boldsymbol{\theta}^*\|_2^2 \\
&\leq \frac{\|\boldsymbol{b}\|_2^2}{\lambda n^2} + \lambda \|\boldsymbol{\theta}^*\|_2^2,
\end{aligned}
$$

which holds for any dataset $\mathcal{D}$, noise $\boldsymbol{b}$, and bandwidth $\mu$. Therefore, taking expectation on both sides gives

$$
\mathbb{E}_{\mathcal{D}, \mathcal{A}} \left[ \widehat{F}_\mu(\widehat{\boldsymbol{\theta}}^{\mathcal{A}}) - \widehat{F}_\mu(\boldsymbol{\theta}^*) \right] \leq \frac{\mathbb{E}\left[ \|\boldsymbol{b}\|_2^2 \right]}{\lambda n^2} + \lambda \|\boldsymbol{\theta}^*\|_2^2 \leq \frac{d\sigma^2}{\lambda n^2} + \lambda \mathrm{R}^2. \tag{23}
$$

The last inequality is by assumption that $\boldsymbol{\theta}^*$ is in $\mathcal{B}(\mathrm{R})$.

Last, plugging Eqs.(22) and (23) back into Eq(21), and setting $\lambda = \sqrt{4C^2/n + d\sigma^2/n^2}/R$, where $\sigma^2 = C^2 \cdot (8\ln(1/\delta) + 8\varepsilon)/\varepsilon^2$ and $C := \sqrt{m\overline{A}^2 + L_h^2 D^2}$, we can get

$$\mathcal{R}(\mathcal{A}; \mathbb{P}) \leq 2R\sqrt{\frac{4C^2}{n} + \frac{d\sigma^2}{n^2}} + [F_\mu(\boldsymbol{\theta}^*) - F(\boldsymbol{\theta}^*)]$$

$$\leq 4\sqrt{2}CR\sqrt{\frac{1}{n} + \frac{d\ln(1/\delta)}{n^2\varepsilon^2}} + [F_\mu(\boldsymbol{\theta}^*) - F(\boldsymbol{\theta}^*)].$$

$\square$

# B  OMITTED MATERIALS AND PROOFS FOR SECTION 4

## B.1  PROOF OF LEMMA 4.2: MONOTONICITY OF $\mathcal{V}_\mu$

*Proof.* For notational convenience, we fix an $\boldsymbol{x}$ and omit the dependency on $\boldsymbol{x}$ in expressions, and denote set $\mathcal{V}_\mu := \mathcal{V}_\mu(\boldsymbol{x}) = \{\boldsymbol{v} \in \mathbb{R}^m : g(\boldsymbol{x} + \mu\boldsymbol{v}) + g(\boldsymbol{x} - \mu\boldsymbol{v}) > 2g(\boldsymbol{x})\}$. First of all, when $\mu = 0$, $\mathcal{V}_0 = \emptyset$. When $\mu = \infty$, because we assume $g$ is not a linear function, then $\mathcal{V}_\infty = \mathbb{R}^m \setminus \{\boldsymbol{0}\}$ due to convexity of $g$. Second, because of convexity of $g$ and Jensen's inequality, the set $\mathcal{V}_\mu \neq \emptyset$ as long as $\mu > 0$.

Now, we come to prove monotonicity. For any $\mu_0 > 0$, suppose we have the set $\mathcal{V}_{\mu_0}$ at hand. Then, for any $\boldsymbol{v} \in \mathcal{V}_{\mu_0}$, by definition of $\mathcal{V}_{\mu_0}$, we must have

$$2g(\boldsymbol{x}) < g(\boldsymbol{x} + \mu_0\boldsymbol{v}) + g(\boldsymbol{x} - \mu_0\boldsymbol{v}), \quad \forall \boldsymbol{v} \in \mathcal{V}_{\mu_0}. \tag{24}$$

Again, because of convexity of $g$, we have:

$$g(\boldsymbol{x} + \mu_0\boldsymbol{v}) \leq g(\boldsymbol{x}) + \langle \nabla g(\boldsymbol{x} + \mu_0\boldsymbol{v}), \mu_0\boldsymbol{v} \rangle, \quad \forall \boldsymbol{v};$$
$$g(\boldsymbol{x} - \mu_0\boldsymbol{v}) \leq g(\boldsymbol{x}) - \langle \nabla g(\boldsymbol{x} - \mu_0\boldsymbol{v}), \mu_0\boldsymbol{v} \rangle, \quad \forall \boldsymbol{v}.$$

Plugging the preceding two inequalities into inequality (24) gives

$$0 < \langle \nabla g(\boldsymbol{x} + \mu_0\boldsymbol{v}) - \nabla g(\boldsymbol{x} - \mu_0\boldsymbol{v}), \boldsymbol{v} \rangle, \quad \forall \boldsymbol{v} \in \mathcal{V}_{\mu_0}. \tag{25}$$

We would like to highlight that the above inequality is strict.

Let $\mu_1 > \mu_0$. We first show the weak version $\mathcal{V}_{\mu_0} \subseteq \mathcal{V}_{\mu_1}$. It suffices to show every $\boldsymbol{v} \in \mathcal{V}_{\mu_0}$ is also in $\mathcal{V}_{\mu_1}$, i.e. $g(\boldsymbol{x} + \mu_1\boldsymbol{v}) + g(\boldsymbol{x} - \mu_1\boldsymbol{v}) > 2g(\boldsymbol{x}), \forall \boldsymbol{v} \in \mathcal{V}_{\mu_0}$. We start with the l.h.s.:

$$\begin{aligned}
g(\boldsymbol{x} + \mu_1\boldsymbol{v}) + g(\boldsymbol{x} - \mu_1\boldsymbol{v}) &= g(\boldsymbol{x} + \mu_0 + (\mu_1 - \mu_0)\boldsymbol{v}) + g(\boldsymbol{x} - \mu_0 - (\mu_1 - \mu_0)\boldsymbol{v}) \\
&\geq [g(\boldsymbol{x} + \mu_0\boldsymbol{v}) + \langle \nabla g(\boldsymbol{x} + \mu_0\boldsymbol{v}), (\mu_1 - \mu_0)\boldsymbol{v} \rangle] \\
&\quad + [g(\boldsymbol{x} - \mu_0\boldsymbol{v}) + \langle \nabla g(\boldsymbol{x} - \mu_0\boldsymbol{v}), -(\mu_1 - \mu_0)\boldsymbol{v} \rangle] \quad \text{(by convexity of } g) \\
&= [g(\boldsymbol{x} + \mu_0\boldsymbol{v}) + g(\boldsymbol{x} - \mu_0\boldsymbol{v})] \\
&\quad + (\mu_1 - \mu_0) \langle \nabla g(\boldsymbol{x} + \mu_0\boldsymbol{v}) - \nabla g(\boldsymbol{x} - \mu_0\boldsymbol{v}), \boldsymbol{v} \rangle.
\end{aligned}$$

Because $\boldsymbol{v} \in \mathcal{V}_{\mu_0}$, we must have $g(\boldsymbol{x} + \mu_0\boldsymbol{v}) + g(\boldsymbol{x} - \mu_0\boldsymbol{v}) > 2f(\boldsymbol{x})$; moreover, inequality (25) implies $(\mu_1 - \mu_0) \langle \nabla g(\boldsymbol{x} + \mu_0\boldsymbol{v}) - \nabla g(\boldsymbol{x} - \mu_0\boldsymbol{v}), \boldsymbol{v} \rangle > 0$. With these two facts, we immediately conclude $g(\boldsymbol{x} + \mu_1\boldsymbol{v}) + g(\boldsymbol{x} - \mu_0\boldsymbol{v}) > 2g(\boldsymbol{x}), \forall \boldsymbol{v} \in \mathcal{V}_{\mu_0}$; thus $\mathcal{V}_{\mu_0} \subseteq \mathcal{V}_{\mu_1}$.

Now, we move to prove the strict subset claim when $g : \boldsymbol{x} \mapsto \|\boldsymbol{x}\|_1$ is the $\ell_1$-norm function. Let us pick the point $\lambda\boldsymbol{v}$ for some $\boldsymbol{v} \in \mathcal{V}_{\mu_0}$ and $\lambda \in (\mu_0/\mu_1, 1)$. It remains to show this point is in $\mathcal{V}_{\mu_1}$ but not in $\mathcal{V}_{\mu_0}$.

- $\lambda\boldsymbol{v}$ **in** $\mathcal{V}_{\mu_1}$: It suffices to show

$$g(\boldsymbol{x} + \mu_1 \cdot \lambda\boldsymbol{v}) + g(\boldsymbol{x} - \mu_1 \cdot \lambda\boldsymbol{v}) \geq \underbrace{g(\boldsymbol{x} + \mu_0\boldsymbol{v}) + g(\boldsymbol{x} - \mu_0\boldsymbol{v})}_{>2f(\boldsymbol{x}), \text{ by } \boldsymbol{v} \in \mathcal{V}_{\mu_0}}$$

$$+ \underbrace{\langle \nabla g(\boldsymbol{x} + \mu_0\boldsymbol{v}) - \nabla g(\boldsymbol{x} - \mu_0\boldsymbol{v}), \mu_1\lambda - \mu_0\boldsymbol{v} \rangle}_{>0, \text{ by inequality (25) and } \lambda > \mu_0/\mu_1}$$

$$> 2g(\boldsymbol{x}),$$

where the second line is by convexity of $g$.

- $\lambda\boldsymbol{v}$ **not in** $\mathcal{V}_{\mu_0}$: We prove this by contradiction. Assume $\lambda\boldsymbol{v}\in\mathcal{V}_{\mu_0}$, then we must have $g(\boldsymbol{x}+\mu_0\lambda\boldsymbol{v})+g(\boldsymbol{x}-\mu_0\lambda\boldsymbol{v})>2g(\boldsymbol{x})$. However, this inequality is not true with $g^{\ell_1}:\boldsymbol{x}\mapsto\|\boldsymbol{x}\|_1$ and $\boldsymbol{x}\in\partial g^{\ell_1}$, because $\|\boldsymbol{x}+\mu_0\lambda\boldsymbol{v}\|_1+\|\boldsymbol{x}-\mu_0\lambda\boldsymbol{v}\|_1=2\|\boldsymbol{x}\|_1=2g^{\ell_1}(\boldsymbol{x}),\forall\boldsymbol{x}\in\partial g^{\ell_1}$. The preceding equality can be verified through the same idea as that in the proof of Lemma 4.1.

Combining both cases, we know the strict subset claim holds for $g^{\ell_1}:\boldsymbol{x}\mapsto\|\boldsymbol{x}\|_1$.

$\square$

### B.2 PROOF OF LEMMA 4.3: KERNEL-DEPENDENT APPROXIMATION ERROR

*Proof.* 1. When kernel function $k(\cdot)$ is Gaussian kernel $k(\boldsymbol{v}):=e^{-\frac{\|\boldsymbol{v}\|_2^2}{2}}/(\sqrt{2\pi})^m$, we have

$$
\begin{aligned}
g_\mu^{\ell_1}(\boldsymbol{x})-g^{\ell_1}(\boldsymbol{x}) &= \int_{\mathcal{V}_\mu^{\ell_1}(\boldsymbol{x})}\left(\frac{\|\boldsymbol{x}+\mu\boldsymbol{v}\|_1+\|\boldsymbol{x}-\mu\boldsymbol{v}\|_1}{2}-\|\boldsymbol{x}\|_1\right)k(\boldsymbol{v})\,d\boldsymbol{v} \\
&\leq \mu\cdot\int_{\boldsymbol{v}:|\boldsymbol{v}|>|\boldsymbol{x}|/\mu}\|\boldsymbol{v}\|_1\,e^{-\frac{\|\boldsymbol{v}\|_2^2}{2}}/(\sqrt{2\pi})^m\,d\boldsymbol{v} && \text{(by Lip cts.)}\\
&\leq \mu\cdot\sum_{j=1}^m\int_{v_j:|v_j|>|x_j|/\mu}|v_j|\,e^{v_j^2/2}/\sqrt{2\pi}\,dv && \text{(layer-by-layer)}\\
&= \sqrt{2/\pi}\mu\cdot\sum_{j=1}^m\exp\left(-\frac{|x_j|^2}{2\mu^2}\right).
\end{aligned}
$$

2. Suppose kernel is exponential kernel $k(\boldsymbol{v})=\frac{1}{\mathfrak{n}}\cdot e^{-\|\boldsymbol{v}\|_2}$ with $\mathfrak{n}=\frac{2\pi^{m/2}\Gamma(m)}{\Gamma(m/2)}$. Let $\|\boldsymbol{x}\|_{-\infty}:=\min_j\{|x_j|\}$ be the minimal element of $|\boldsymbol{x}|$, and let $S(\cdot)$ be the surface measure for any given set. Then, we have,

$$
\begin{aligned}
g_\mu^{\ell_1}(\boldsymbol{x})-g^{\ell_1}(\boldsymbol{x}) &\leq \mu\cdot\int_{\boldsymbol{v}:|\boldsymbol{v}|>|\boldsymbol{x}|/\mu}\|\boldsymbol{v}\|_1\frac{e^{-\|\boldsymbol{v}\|_2}}{\mathfrak{n}}\,d\boldsymbol{v} \\
&\leq \mu\sqrt{m}\cdot\int_{\boldsymbol{v}:|\boldsymbol{v}|>|\boldsymbol{x}|/\mu}\|\boldsymbol{v}\|_2\frac{e^{-\|\boldsymbol{v}\|_2}}{\mathfrak{n}}\,d\boldsymbol{v} \\
&= \frac{\mu\sqrt{m}}{\mathfrak{n}}\int_{\|\boldsymbol{x}\|_{-\infty}/\mu}^\infty te^{-t}\cdot\underbrace{S(\mathcal{B}_2(t))}_{=2\pi^{m/2}t^{m-1}/\Gamma(m/2)}\,dt && \text{(volumn of a ball)}\\
&= \frac{\mu\sqrt{m}}{\Gamma(m)}\cdot\underbrace{\int_{\|\boldsymbol{x}\|_{-\infty}/\mu}^\infty t^m e^{-t}\,dt}_{=:\Gamma(m+1,\|\boldsymbol{x}\|_{-\infty}/\mu)}, && (\Gamma(\cdot)\text{ is Gamma function})
\end{aligned}
$$

$$(26)$$

where the function $\Gamma(m+1,r):=\int_r^\infty t^m e^{-t}\,dt$ in (26) is the upper incomplete gamma function. The upper incomplete gamma function has a closed-form when $m$ is positive integer, i.e., $\Gamma(m+1,r)=m!\cdot e^{-r}\cdot\sum_{k=0}^m\frac{r^k}{k!}$. Moreover, the upper incomplete gamma function has a light tail; that is, when $r$ increases, the function value tends to zero very fast. The rate can be characterized if we can find a tight upper bound for the series $\sum_{k=0}^m\frac{r^k}{k!}$.

We notice that, when $r \geq 3$:

$$
\begin{aligned}
\ln\left(\sum_{k=0}^{m} \frac{r^k}{k!}\right) = \ln\left(\sum_{k=0}^{m} \frac{r^k}{k!}\right) &\leq \ln\left(\sum_{k=0}^{m} \frac{r^k}{e \cdot k^k/e^k}\right) && \text{(by Stirling's approximation)} \\
&\leq \ln\left(\frac{1}{e} \cdot \sum_{k=0}^{m}(re)^k\right) \\
&= \ln\left(\frac{1}{e} \cdot \frac{(re)^{m+1}-1}{re-1}\right) \\
&\leq (m+1) \cdot (\ln r + 1) - \ln(re-1) - 1 \\
&= m \cdot (\ln r + 1) + \ln\left(\frac{r}{re-1}\right) \\
&\leq 2m \ln r. && \text{(by } r \geq 3)
\end{aligned}
$$

Consequently, an upper bound for the upper incomplete gamma function follows:

$$
\Gamma(m+1, r) \leq m! \cdot e^{-r} \cdot e^{2m \ln r} = m! \cdot e^{-r} \cdot r^{2m}, \quad \forall r \geq 3.
$$

Immediately,

$$
g_\mu^{\ell_1}(\boldsymbol{x}) - g^{\ell_1}(\boldsymbol{x}) \leq m^{3/2} \cdot \mu \cdot \exp\left(-\frac{\|\boldsymbol{x}\|_{-\infty}}{\mu} + 2m \ln\left(\frac{\|\boldsymbol{x}\|_{-\infty}}{\mu}\right)\right).
$$

3. When kernel is Laplacian kernel $k(\boldsymbol{v}) = \frac{1}{2^m} \cdot e^{-\|\boldsymbol{v}\|_1}$, we have

$$
\begin{aligned}
g_\mu^{\ell_1}(\boldsymbol{x}) - g^{\ell_1}(\boldsymbol{x}) &\leq \mu \cdot \int_{\boldsymbol{v}:|\boldsymbol{v}|>|\boldsymbol{x}|/\mu} \left(\sum_{j=1}^{m}|v_j|\right) \cdot \frac{1}{2^m} \cdot e^{-\|\boldsymbol{v}\|_1} \, d\boldsymbol{v} \\
&\leq \mu \cdot \sum_{j=1}^{m}\left[\int_{v_j:|v_j|>|x_j|/\mu} |v_j| \cdot e^{-|v_j|}/2 \, d\boldsymbol{v}\right] \\
&\leq \mu \cdot \sum_{j=1}^{m}\left(\frac{|x_j|}{\mu} + 1\right) \exp\left(-\frac{|x_j|}{\mu}\right).
\end{aligned}
$$

$\square$

## B.3 PROOF OF THEOREM 4.5: OP'S OPTIMAL RATES

*Proof.* Because we use Gaussian kernel, by Lemmas 3.3 and 4.3, we know the excess generalization risk can be upper bounded as

$$
\mathcal{R}(\mathcal{A}; \mathbb{P}) \leq 4\sqrt{2}CR \cdot \left(\frac{1}{\sqrt{n}} + \frac{\sqrt{d\ln(1/\delta)}}{n\varepsilon}\right) + \sqrt{2/\pi}\mu \cdot \mathbb{E}_{\boldsymbol{z}}\left[\sum_{j=1}^{m}\exp\left(-\frac{|A(\boldsymbol{z})\boldsymbol{\theta}^*|_j^2}{2\mu^2}\right)\right]. \quad (27)
$$

It remains to check the upper bound when the value $\mu$ is chosen according to the algorithm C-OP in Step 1, i.e., $\lambda = \frac{(\beta_\mu + \beta_h)(m+1)}{n\varepsilon}$. Moreover, Table 1 indicates $\beta_\mu := L_f/\mu$, where $L_f := \sqrt{m}\overline{A} + DL_h$. Furthermore, by Lemma 3.3, we set $\lambda := \sqrt{4C^2/n + d\sigma^2/n^2}/R$ with $C := \sqrt{m\overline{A}^2 + D^2 L_h^2}$ and $\sigma^2 = C^2 \cdot (8\ln(1/\delta) + 8\varepsilon)/\varepsilon^2$. Therefore, combining these relationships together, we can explicitly find a $\mu$ used by C-OP; specifically, the $\mu$ used is $\mu := \frac{(m+1)L_f}{2CR\sqrt{n\varepsilon^2 + 2d(\ln(1/\delta) + \varepsilon)} - \beta_h}$.

Now, we come to control pointwise approximation error upper bound. If Gaussian kernel is used, then Lemma 4.3 implies, by law of total probability,

$$
\mathbb{E}_{\boldsymbol{z}}\left[\sum_{j=1}^{m}\exp\left(-\frac{|A(\boldsymbol{z})\boldsymbol{\theta}^*|_j^2}{2\mu^2}\right)\right] = \mathbb{E}_{\boldsymbol{z}}\left[\sum_{j=1}^{m}\exp\left(-\frac{|A(\boldsymbol{z})\boldsymbol{\theta}^*|_j^2}{2\mu^2}\right)\mid \|A(\boldsymbol{z})\boldsymbol{\theta}^*\|_{-\infty}\geq t\right]\cdot p(t)
$$

$$
+ \mathbb{E}_{\boldsymbol{z}}\left[\sum_{j=1}^{m}\exp\left(-\frac{|A(\boldsymbol{z})\boldsymbol{\theta}^*|_j^2}{2\mu^2}\right)\mid \|A(\boldsymbol{z})\boldsymbol{\theta}^*\|_{-\infty}< t\right]\cdot(1-p(t))
$$

$$
\leq m\exp\left(-t^2/(2\mu^2)\right) + m\cdot(1-p(t))
$$

$$
= m\left(\exp\left(-t^2/(2\mu^2)\right) + 1 - p(t)\right). \tag{28}
$$

Since assumption 4.4 says $p(t)\geq 1-\exp\left(-1/t^2\right), \forall t\leq \tau$, if we take $t=\sqrt{\mu}$, then Eq.(28) becomes

$$
(28) = m\left(\exp\left(-1/(2\mu)\right) + \exp\left(-1/\mu\right)\right) \leq 2m\exp\left(-1/\mu\right). \tag{29}
$$

The preceding inequality holds only when $\sqrt{\mu}$ is smaller than the threshold $\tau$. Nevertheless, we will see later that the chosen $\mu$ tends to 0 as sample size $n$ goes to infinity. Substituting (29) into (27), we get the final utility guarantee:

$$
\mathcal{R}(\mathcal{A};\mathbb{P}) \leq 4\sqrt{2}CR\cdot\left(\frac{1}{\sqrt{n}} + \frac{\sqrt{d\ln(1/\delta)}}{n\varepsilon}\right) + 2\sqrt{2}m\mu\exp\left(-1/\mu\right).
$$

Intuitively, if $\mu := \frac{(m+1)L_f}{2CR\sqrt{n\varepsilon^2+2d(\ln(1/\delta)+\varepsilon)}-\beta_h} \searrow 0$ as $n\to\infty$, then the approximation error $2\sqrt{2}m\mu\exp(-1/\mu)$, which decreases exponentially in $\mu$, will be dominated by the polynomial term $4\sqrt{2}CR\left(\frac{1}{\sqrt{n}} + \frac{\sqrt{d\ln(1/\delta)}}{n\varepsilon}\right)$. It is straightforward to find out sufficient conditions for

$$
\begin{cases}
\mu \leq \tau^2 \\
2\sqrt{2}m\mu\exp\left(-1/\mu\right) \leq 4\sqrt{2}CR\left(\frac{1}{\sqrt{n}} + \frac{\sqrt{d\ln(1/\delta)}}{n\varepsilon}\right)
\end{cases}
$$

$$
\Leftarrow \quad \mu \leq \min\left\{1, \tau^2, 1/\ln\left(\frac{m}{2CR(\frac{1}{\sqrt{n}} + \frac{\sqrt{d\ln(1/\delta)}}{n\varepsilon})}\right)\right\}
$$

$$
\Leftarrow \quad \begin{cases}
2CR\sqrt{n\varepsilon^2+2d(\ln(1/\delta)+\varepsilon)} \geq \beta_h \\
2CR\sqrt{n\varepsilon^2+2d(\ln(1/\delta)+\varepsilon)} \geq \beta_h + (m+1)L_f\cdot\max\left\{1, 1/\tau^2, \ln\left(\frac{m}{2CR(\frac{1}{\sqrt{n}} + \frac{\sqrt{d\ln(1/\delta)}}{n\varepsilon})}\right)\right\}
\end{cases}
$$

$$
\Leftarrow \quad \begin{cases}
n\varepsilon^2+2d(\ln(1/\delta)+\varepsilon) \geq \frac{\beta_h^2}{C^2R^2}; \\
n\varepsilon^2+2d(\ln(1/\delta)+\varepsilon) \geq \frac{(m+1)^2L_f^2}{2C^2R^2}\cdot\min\left\{\ln\left(\frac{m\sqrt{n}}{2CR}\right)^2, \ln\left(\frac{mn\varepsilon}{2CR}\right)^2\right\}; \\
n\varepsilon^2+2d(\ln(1/\delta)+\varepsilon) \geq \frac{(m+1)^2L_f^2/\min\{1,\tau^4\}}{C^2R^2}.
\end{cases} \tag{30}
$$

The first inequality in Eq.(30) ensures that the denominator in the expression of $\mu$ is always positive. Below are some sufficient conditions ensuring the above inequalities to be true.

1. $\delta \leq \min\left\{e^{-\frac{\max\{\beta_h^2,(m+1)^2L_f^2/\min\{1,\tau^4\}\}}{dC^2R^2}}, \left(\frac{2CR}{m\sqrt{n}}\right)^{(m+1)^2/(2dR^2)}\right\}$. First of all, $\delta \leq e^{-\frac{\max\{\beta_h^2,(m+1)^2L_f^2/\min\{1,\tau^4\}\}}{dC^2R^2}}$ is equivalent to $d\ln(1/\delta)\geq \frac{\max\{\beta_h^2,(m+1)^2L_f^2/\min\{1,\tau^4\}\}}{C^2R^2}$, which implies the first and last inequalities of (30).

   Next, when $\delta \leq \left(\frac{2CR}{m\sqrt{n}}\right)^{(m+1)^2/(2dR^2)}$, we have $\delta \leq \left(\frac{2CR}{m\sqrt{n}}\right)^{(m+1)^2/(2dR^2)} \leq \left(\frac{2CR}{m\sqrt{n}}\right)^{(m+1)^2L_f^2/(2dC^2R^2)} = \exp\left(\frac{-(m+1)^2L_f^2}{2dC^2R^2}\cdot\ln\left(\frac{m\sqrt{n}}{2CR}\right)\right)$, which implies $d\ln(1/\delta)\geq \frac{(m+1)^2L_f^2}{2C^2R^2}\cdot\ln\left(\frac{m\sqrt{n}}{2CR}\right)$.

Combining both cases, we know the $\delta$ chosen is sufficient for Eq.(30).

2. $\delta \leq e^{-\frac{\max\{\beta_h^2 \cdot (m+1)^2 L_f^2 / \min\{1, \tau^4\}\}}{2dC^2 R^2}}$ and $\varepsilon \geq \sqrt{\frac{2(m+1)^2 \ln\left(\frac{m\sqrt{n}}{2CR}\right)}{nR^2}}$. We only need to check the condition on $\varepsilon$. Direct calculation gives $n\varepsilon^2 \geq \frac{2(m+1)^2}{R^2} \ln\left(\frac{m\sqrt{n}}{2CR}\right) \geq \frac{2(m+1)^2 \cdot L_f^2}{2C^2 R^2} \ln\left(\frac{m\sqrt{n}}{2CR}\right) = \frac{(m+1)^2 L_f^2}{C^2 R^2} \ln\left(\frac{m\sqrt{n}}{2CR}\right)$.

$\square$

## B.4 OMITTED DISCUSSIONS ON SOME SPECIAL CASES

- **Piecewise Linear Loss** $f(\boldsymbol{\theta}; \boldsymbol{z}) := \max_{p \in [P]}\{\langle \boldsymbol{a}_p, A(\boldsymbol{z})\boldsymbol{\theta}\rangle + b_p\}$. The piecewise linear loss is essentially a linear model. The privacy guarantee follows the same proof idea as in the proof of Theorem 3.2, with slight modifications. First, the piecewise linear loss does not have an $h(\cdot)$ function; so $h(\cdot)$ related terms, such as $D^2 L_h^2$, can be removed. Second, the additional term $\boldsymbol{a}_p$ makes the variance in Eq.(16) larger by at most $\sup_{p \in [P]} \|\boldsymbol{a}_p\|_2^2$. Therefore, the multiplicative constant $C$ in the noise variance now should be $C^2 := m\overline{A}^2 \cdot \sup_{p \in [P]} \|\boldsymbol{a}_p\|_2^2$.

  To obtain the optimal convergence rate, we need a analogue of Assumption 4.4 to ensure $A(z)\boldsymbol{\theta}^*$ stays away from the set of critical points $\mathcal{Z} := \{\boldsymbol{z} \in \mathcal{Z} : \exists i, j \in [P], s.t. \langle \boldsymbol{a}_i, A(\boldsymbol{z})\boldsymbol{\theta}^*\rangle = \langle \boldsymbol{a}_j, A(\boldsymbol{z})\boldsymbol{\theta}^*\rangle\}$. The set $\mathcal{Z}$ contains all $\boldsymbol{z}'s$ where there are at least two pieces intersects with each other. Let $r(\boldsymbol{z}, \mathcal{Z}) := \min_{\boldsymbol{z}_0 \in \mathcal{Z}}\{\|\boldsymbol{z} - \boldsymbol{z}_0\|_{-\infty}\}$ be the distance between point $\boldsymbol{z}$ and set $\mathcal{Z}$, where $\|\boldsymbol{z}\|_{-\infty} := \min\{|z_1|, \ldots, |z_d|\}$ is the minimal absolute value of vector $\boldsymbol{z}$.

  Another key ingredient is to ensure the set defined follow is strictly monotone in $\mu$,

$$\mathcal{V}_\mu(\boldsymbol{z}) := \left\{\boldsymbol{v} \subseteq \mathbb{R}^m : \max_{p \in [P]}\{\langle \boldsymbol{a}_p, A(\boldsymbol{z})\boldsymbol{\theta} + \mu\boldsymbol{v}\rangle + b_p\} + \max_{p \in [P]}\{\langle \boldsymbol{a}_p, A(\boldsymbol{z})\boldsymbol{\theta} - \mu\boldsymbol{v}\rangle + b_p\} \right.$$
$$\left. > 2\max_{p \in [P]}\{\langle \boldsymbol{a}_p, A(\boldsymbol{z})\boldsymbol{\theta}\rangle + b_p\}\right\},$$

  on which the corresponding integrand is strictly positive. However, the set $\mathcal{V}_\mu(\boldsymbol{z})$ is hard to characterize. But we can find a superset of it:

$$\overline{\mathcal{V}}_\mu(\boldsymbol{z}) := \{\boldsymbol{v} \in \mathbb{R}^m : \|\boldsymbol{v}\|_\infty > r(\boldsymbol{z}, \mathcal{Z})/\mu\}.$$

  Because for any $\boldsymbol{v}$ s.t. $\|\boldsymbol{v}\|_\infty \leq r(\boldsymbol{z}, \mathcal{Z})/\mu$, the value $A(\boldsymbol{z})\boldsymbol{\theta} \pm \mu\boldsymbol{v}$ does not move away from the piece where $A(\boldsymbol{z})\boldsymbol{\theta}$ originally lives in, we can immediately conclude that such a $\boldsymbol{v} \notin \mathcal{V}_\mu(\boldsymbol{z})$. By a contrapositive argument, when $\boldsymbol{v} \in \mathcal{V}_\mu(\boldsymbol{z})$, then $\|\boldsymbol{v}\|_\infty \not\leq r(\boldsymbol{z}, \mathcal{Z})/\mu$, i.e. $\boldsymbol{v} \in \overline{\mathcal{V}}_\mu(\boldsymbol{z})$. Therefore, $\mathcal{V}_\mu(\boldsymbol{z}) \subseteq \overline{\mathcal{V}}_\mu(\boldsymbol{z})$.

  **Assumption B.1.** Let $\boldsymbol{z} \sim \mathbb{P}$. We assume there exists a threshold $\tau > 0$ such that $\mathbb{P}_{\boldsymbol{z}}[r(\boldsymbol{z}, \mathcal{Z}) \geq t] \geq 1 - \exp\left(-1/t^2\right), \forall t \leq \tau$.

  With this assumption and assumption 2.3, proof in Appendix B.3 then can go through for $\overline{\mathcal{V}}_\mu(\boldsymbol{z})$.

- **Bowl-shaped Loss** $f(\boldsymbol{\theta}; z) := (\|A\boldsymbol{\theta}\|_1 - z)^+$ with known $A \in \mathbb{R}^{m \times d}$. Because the positive operator has Lipschitz constant 1, this operator will not affect noise magnitude; thus if we let the constant $C^2 = m\overline{A}^2$ in $\sigma^2$, then $(\varepsilon, \delta)$-DP is guaranteed. Regarding convergence rates, we first let $\boldsymbol{x}^* := A\boldsymbol{\theta}^*$. Similarly, we denote

$$\mathcal{V}_\mu(z) := \{\boldsymbol{v} \in \mathbb{R}^d : (\|\boldsymbol{x}^* + \mu\boldsymbol{v}\|_1 - z)^+ + (\|\boldsymbol{x}^* - \mu\boldsymbol{v}\|_1 - z)^+ - 2(\|\boldsymbol{x}^*\|_1 - z)^+ > 0\}.$$

  It can be shown that $\mathcal{V}_\mu(z) \subseteq S_\mu(\boldsymbol{z})$, where $S_\mu(z)$'s complementary set $S^C(z) := \left\{\boldsymbol{v} \in \mathbb{R}^d \middle| \begin{array}{l} \mu|v_i| \leq |x_i^*|, \forall i = 1, \ldots, m; \\ \mu\|\boldsymbol{v}\|_1 \leq \|\|\boldsymbol{x}^*\|_1 - z\| \end{array}\right\}$. The set $S_\mu^C(z)$ is highlighted in blue in Figure 6). For any $\boldsymbol{v} \in S_\mu^C(z)$, it must be true that $\mathsf{sgn}(\boldsymbol{x}^*) = \mathsf{sgn}(\boldsymbol{x}^* + \mu\boldsymbol{v}) = \mathsf{sgn}(\boldsymbol{x}^* - \mu\boldsymbol{v})$,

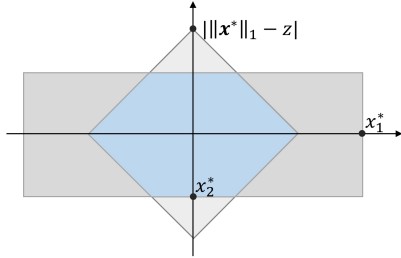

Figure 6: Set $S_\mu^C(z)$ is in blue (when $\mu = 1$)

which follows from the first constraint in $S_\mu^C(z)$. An immediate consequence of the unchanged signs is

$$\|\boldsymbol{x}^* + \mu\boldsymbol{v}\|_1 + \|\boldsymbol{x}^* - \mu\boldsymbol{v}\|_1 = 2\|\boldsymbol{x}^*\|_1. \tag{31}$$

The second constraint in $S_\mu^C(z)$ implies

$$\left|\sum_{i=1}^m \mathsf{sgn}(x_i^*)\mu v_i\right| \leq \sum_{i=1}^m |\mathsf{sgn}(x_i^*)\mu v_i| = \mu\|\boldsymbol{v}\|_1 \leq |\|\boldsymbol{x}^*\|_1 - z|$$

$$\implies \left|\|\boldsymbol{x}^*\|_1 - z + \sum_{i=1}^m \mathsf{sgn}(x_i^*)\mu v_i\right| + \left|\|\boldsymbol{x}^*\|_1 - z + \sum_{i=1}^m \mathsf{sgn}(x_i^*) \cdot (-\mu v_i)\right| = 2|\|\boldsymbol{x}^*\|_1 - z|$$

$$\iff \left|\sum_{i=1}^m \{\mathsf{sgn}(x_i^*) \cdot (x_i^* + \mu v_i)\} - z\right| + \left|\sum_{i=1}^m \{\mathsf{sgn}(x_i^*) \cdot (x_i^* - \mu v_i)\} - z\right| = 2|\|\boldsymbol{x}^*\|_1 - z|$$

$$\iff \left|\sum_{i=1}^m \{\mathsf{sgn}(x_i^* + \mu v_i) \cdot (x_i^* + \mu v_i)\} - z\right| + \left|\sum_{i=1}^m \{\mathsf{sgn}(x_i^* - \mu v_i) \cdot (x_i^* - \mu v_i)\} - z\right| = 2|\|\boldsymbol{x}^*\|_1 - z|$$

$$\iff |\|\boldsymbol{x}^* + \mu\boldsymbol{v}\|_1 - z| + |\|\boldsymbol{x}^* - \mu\boldsymbol{v}\|_1 - z| = 2|\|\boldsymbol{x}^*\|_1 - z|. \tag{32}$$

With these two Eqs.(31) and (32), we are ready to show $S_\mu^C(z) \subseteq \mathcal{V}_\mu^C(z)$: for any $\boldsymbol{v} \in S_\mu^C(z)$, we have:

$$(\|\boldsymbol{x}^* + \mu\boldsymbol{v}\|_1 - z)^+ + (\|\boldsymbol{x}^* - \mu\boldsymbol{v}\|_1 - z)^+$$

$$-2(\|\boldsymbol{x}^*\|_1 - z)^+ = \frac{|\|\boldsymbol{x}^* + \mu\boldsymbol{x}\|_1 - z| + (\|\boldsymbol{x}^* + \mu\boldsymbol{v}\|_1 - z)}{2}$$

$$+ \frac{|\|\boldsymbol{x} - \mu\boldsymbol{v}\|_1 - z| + (\|\boldsymbol{x}^* - \mu\boldsymbol{v}\|_1 - z)}{2}$$

$$- \frac{2(|\|\boldsymbol{x}^*\|_1 - z| + (\|\boldsymbol{x}^*\|_1 - z))}{2}$$

$$= \frac{\|\boldsymbol{x}^* + \mu\boldsymbol{v}\|_1 + \|\boldsymbol{x}^* - \mu\boldsymbol{v}\|_1 - 2\|\boldsymbol{x}^*\|_1}{2}$$

$$+ \frac{|\|\boldsymbol{x}^* + \mu\boldsymbol{v}\|_1 - z| + |\|\boldsymbol{x}^* - \mu\boldsymbol{v}\|_1 - z| - 2|\|\boldsymbol{x}^*\|_1 - z|}{2}$$

$$= 0 + 0 = 0,$$

where the last line is by Eq.(31) and Eq.(32). The above derivation implies $\boldsymbol{v} \in \mathcal{V}_\mu^C(z)$. Therefore $S_\mu^C(z) \subseteq \mathcal{V}_\mu^C(z)$, and consequently, $\mathcal{V}_\mu(z) \subset S_\mu(z)$.

We next assume the distribution of $z$ is not ill-posed.

**Assumption B.2.** Let $z \sim \mathbb{P}$. We assume there exists a threshold $\tau > 0$ such that $\mathbb{P}_z[|z - \|A\boldsymbol{\theta}^*\|_1| \geq t] \geq 1 - \exp(-1/t^2), \forall t \leq \tau$.

Then, with this assumption and assumption 2.3, similarly, proof in Appendix B.3 then can go through for $S_\mu(z)$.

B.5   PROOF OF LEMMA 4.6: PROPERTIES OF MOREAU ENVELOPE

*Proof.* We first introduce two Lemmas on *Fenchel's conjugate* that are helpful in later analysis. For a function $f$, denote its Fenchel Conjugate as $f^\star(\boldsymbol{y}) := \sup_{\boldsymbol{x} \in dom f} \langle \boldsymbol{x}, \boldsymbol{y} \rangle - f(\boldsymbol{x})$.

**Lemma B.3** (Fenchel's Duality, Theorem 31.1 in Rockafellar 2015). *Let $f, h$ be two proper convex functions. If the intersection between the relative interior of domains of functions $f$ and $g$ are nonempty, i.e., $ri(dom f) \cap ri(dom h) \neq \emptyset$, then one has*

$$\inf_{\boldsymbol{u}} \{f(\boldsymbol{u}) - h(\boldsymbol{u})\} = \sup_{\boldsymbol{y}} \{h^\star(\boldsymbol{y}) - f^\star(\boldsymbol{y})\}.$$

**Lemma B.4** (Conjugate Correspondence Theorem, Theorem 5.26 in Beck 2017). *Let $f : \mathbb{R}^d \to \mathbb{R}$ be a proper lower semicontinuous convex function. The following statements are equivalent:*

- *function $f$ is $1/\mu$-smooth with respect to $\|\cdot\|$;*

- *its Fenchel conjugate $f^\star$ is $\mu$-strongly convex with respect to dual norm $\|\cdot\|_*$*

Throughout this proof, we replace the subscript $_{\mathsf{ME}}$ with $\mu$ to indicate the dependency on parameter $\mu$. By applying Lemma B.3, we first rewrite the Generalized Moreau Envelope into a dual formulation:

$$g_\mu(\boldsymbol{x}) := \inf_{\boldsymbol{u}} \{g(\boldsymbol{u}) + \phi_\mu(\boldsymbol{x} - \boldsymbol{u})\} = \inf_{\boldsymbol{u}} \{g(\boldsymbol{u}) - h_{\mu,\boldsymbol{x}}(\boldsymbol{u})\} \qquad (\text{let } h_{\mu,\boldsymbol{x}}(\boldsymbol{u}) := -\phi_\mu(\boldsymbol{x} - \boldsymbol{u}))$$

$$= \sup_{\boldsymbol{y}} \{h_{\mu,\boldsymbol{x}}^\star(\boldsymbol{y}) - g^\star(\boldsymbol{y})\}. \qquad (\text{by Lemma B.3})$$

$$(33)$$

Since

$$h_{\mu,\boldsymbol{x}}^\star(\boldsymbol{y}) = \sup_{\mathbf{g}} \langle \mathbf{g}, \boldsymbol{y} \rangle + \phi_\mu(\boldsymbol{x} - \mathbf{g})$$

$$= \langle \boldsymbol{x}, \boldsymbol{y} \rangle - \inf_{\mathbf{g}} \{\langle \boldsymbol{x} - \mathbf{g}, \boldsymbol{y} \rangle - \phi_\mu(\boldsymbol{x} - \mathbf{g})\} \qquad (\text{safe here since } dom\phi_\mu = \mathbb{R}^d)$$

$$= \langle \boldsymbol{x}, \boldsymbol{y} \rangle - \phi_\mu^\star(\boldsymbol{y}),$$

we can plug $h_{\mu,\boldsymbol{x}}^\star = \langle \boldsymbol{x}, \boldsymbol{y} \rangle - \phi_\mu^\star(\boldsymbol{y})$ back into equation 33, and obtain

$$g_\mu(\boldsymbol{x}) = \sup_{\boldsymbol{y}} \{\langle \boldsymbol{x}, \boldsymbol{y} \rangle - \phi_\mu^\star(\boldsymbol{y}) - g^\star(\boldsymbol{y})\} \qquad (34)$$

$$= \sup_{\boldsymbol{y}} \{\langle \boldsymbol{x}, \boldsymbol{y} \rangle - (g^\star + \phi_\mu^\star)(\boldsymbol{y})\} = (g^\star + \phi_\mu^\star)^\star(\boldsymbol{x}). \qquad (35)$$

With the Fenchel dual expression, we are ready to show desired properties.

1. The convexity of $g_\mu$ is straightforward, as $g_\mu$ is obtained from minimizing a convex function over a convex set; thus convexity is preserved.

   The smoothness comes from applying Lemma B.4 to $\phi_\mu$. Specifically, with a properly chosen function $\phi$ that is continuously differentiable and $\beta'$-smooth (w.r.t $\|\cdot\|_2$), we have

   $$\|\nabla\phi_\mu(\boldsymbol{x}) - \nabla\phi_\mu(\boldsymbol{y})\|_q = \|\nabla\phi(\boldsymbol{x}/\mu) - \nabla\phi(\boldsymbol{y}/\mu)\|_q \leq \beta' \cdot \|\boldsymbol{x}/\mu - \boldsymbol{y}/\mu\|_p = \frac{\beta'}{\mu} \cdot \|\boldsymbol{x} - \boldsymbol{y}\|_p,$$

   which implies $\phi_\mu$ is $\beta'/\mu$-smooth (w.r.t $\|\cdot\|_p$). Applying Lemma B.4, we know that $\phi_\mu^\star$ is $\mu/\beta'$-strongly convex (w.r.t $\|\cdot\|_q$). As a result, $g^\star + \phi_\mu^\star$ is also $\mu/\beta'$-strongly convex. Then applying again Lemma B.4 to $g^\star + \phi_\mu^\star$, we have $(g^\star + \phi_\mu^\star)^\star$ is $\beta'/\mu$-smooth (w.r.t $\|\cdot\|_p$). By the equivalence between $(g^\star + \phi_\mu^\star)^\star$ and $g_\mu$, i.e., Eq.(35), we know $g_\mu$ is $\beta := \beta'/\mu$-smooth.

   Lastly, we come to show $g_\mu$ is $L$-Lipschitz. For a given $\boldsymbol{x}$, pick an arbitrary subgradient $\boldsymbol{v} \in \partial g_\mu(\boldsymbol{x})$. We know that Fenchel-Young inequality can characterize subgradients when taking equality, i.e.,

   $$\langle \boldsymbol{v}, \boldsymbol{x} \rangle = g_\mu(\boldsymbol{x}) + g_\mu^\star(\boldsymbol{v}), \quad \forall \boldsymbol{v} \in \partial g_\mu(\boldsymbol{x}). \qquad (36)$$

In addition, because $\phi$ is strictly convex, $u^* := \arg\min_{\boldsymbol{u}} g(\boldsymbol{u}) + \phi_\mu(\boldsymbol{x} - \boldsymbol{u})$ is unique for any $\boldsymbol{x}$; and therefore $f_\mu(\boldsymbol{x})$ can be expressed as $g_\mu(\boldsymbol{x}) = g(u^*) + \phi_\mu(\boldsymbol{x} - \boldsymbol{u}^*)$. Moreover, we have $g_\mu^\star = g^\star + \phi_\mu^\star$ by Eq.(35). Hence, Eq.(36) can be equivalently rewritten as

$$\langle \boldsymbol{v}, \boldsymbol{x} - \boldsymbol{u}^* \rangle + \langle \boldsymbol{v}, \boldsymbol{u}^* \rangle = \text{LHS of (36)}$$
$$= \text{RHS of (36)} = [g(\boldsymbol{u}^*) + \phi_\mu(\boldsymbol{x} - \boldsymbol{u}^*)] + [g^\star(\boldsymbol{v}) + \phi_\mu^\star(\boldsymbol{v})] . \tag{37}$$

Since $\phi_\mu$ is continuously differentiable, its subgradient set is singleton and is $\partial\phi_\mu(\boldsymbol{x} - \boldsymbol{u}^*) = \{\nabla\phi_\mu(\boldsymbol{x} - \boldsymbol{u}^*)\}, \forall \boldsymbol{x}$. By the second property in Lemma 4.6, i.e., $\nabla g_\mu(\boldsymbol{x}) = \nabla\phi_\mu(\boldsymbol{x} - \boldsymbol{u}^*)$, the set of subgradients of $g_\mu(\boldsymbol{x})$ is also singleton since $\partial g_\mu(\boldsymbol{x}) = \partial\phi_\mu(\boldsymbol{x} - \boldsymbol{u}^*) = \{\nabla\phi_\mu(\boldsymbol{x} - \boldsymbol{u}^*)\}$. The coincidence between $\partial g_\mu(\boldsymbol{x})$ and $\partial\phi_\mu(\boldsymbol{x} - \boldsymbol{u}^*)$ implies that the picked subgradient $\boldsymbol{v} \in \partial f_\mu(\boldsymbol{x})$ should be also in $\partial\phi_\mu(\boldsymbol{x} - \boldsymbol{u}^*)$. By Fenchel-Young equality again, we have

$$\langle \boldsymbol{v}, \boldsymbol{x} - \boldsymbol{u}^* \rangle = \phi_\mu(\boldsymbol{x} - \boldsymbol{u}^*) + \phi_\mu^\star(\boldsymbol{v}). \tag{38}$$

Subtracting Eq.(38) from Eq.(37) results in

$$\langle \boldsymbol{v}, \boldsymbol{u}^* \rangle = g(\boldsymbol{u}^*) + g^\star(\boldsymbol{v}),$$

which implies $\boldsymbol{v} \in \partial g(\boldsymbol{u}^*)$. The above analysis shows that the picked subgradient $\boldsymbol{v} \in \partial g_\mu(\boldsymbol{x})$ belongs to two other sets $\partial\phi_\mu(\boldsymbol{x} - \boldsymbol{u}^*)$ and $\partial g(\boldsymbol{u}^*)$ at the same time, implying that $\boldsymbol{v} \in \partial\phi_\mu(\boldsymbol{x} - \boldsymbol{u}^*) \cap \partial g(\boldsymbol{u}^*) \subseteq \partial g(\boldsymbol{u}^*)$. Therefore, the norm of $\boldsymbol{v}$ is upper bounded by the largest norm of subgradients in $\partial g(\boldsymbol{u}^*)$, i.e., :

$$\|\boldsymbol{v}\| \leq \sup_{\mathbf{g} \in \partial g(\boldsymbol{u}^*)} \|\mathbf{g}\| , \quad \forall \boldsymbol{x}, \boldsymbol{v}.$$

kindly note that $\boldsymbol{u}^*$ depends on $\boldsymbol{x}$. By the assumption that $g$ is $L$-Lipschitz continuous w.r.t. $\|\cdot\|_p$, subgradients in $\partial g(\boldsymbol{u}^*)$ are uniformly bounded w.r.t. dual norm $\|\cdot\|_q$, i.e., $\sup_{\mathbf{g} \in \partial g(\boldsymbol{u}^*)} \|\mathbf{g}\|_q \leq L$. When we are in Euclidean space $\|\cdot\|_p = \|\cdot\|_q = \|\cdot\|_2$, we finally have $\sup_{\mathbf{g} \in \partial g(\boldsymbol{u}^*)} \|\mathbf{g}\|_2 \leq L$, which directly leads to

$$\sup_{\boldsymbol{v} \in \partial g_\mu(\boldsymbol{x}), \forall \boldsymbol{x} \in \mathcal{X}} \|\boldsymbol{v}\|_2 \leq \sup_{\mathbf{g} \in \partial g(\boldsymbol{u}^*), \forall \boldsymbol{x}} \|\mathbf{g}\|_2 \leq L,$$

indicating that $g_\mu$ is $L$-Lipschitz continuous.

2. The second property is a well-known result in Moreau Envelope literature, see Theorem 4.1 in Beck & Teboulle (2012). This property also indicates that $g_\mu$ is differentiable.

3. It is easy to see

$$g_\mu(\boldsymbol{x}) - g(\boldsymbol{x}) = \inf_{\boldsymbol{u}} \{g(\boldsymbol{u}) + \phi_\mu(\boldsymbol{x} - \boldsymbol{u})\} - g(\boldsymbol{x}) \leq g(\boldsymbol{x}) + \phi_\mu(\mathbf{0}) - g(\boldsymbol{x}) = \mu\phi(\mathbf{0}), \quad \forall \boldsymbol{x},$$

which gives the upper bound. To show the desired lower bounds, we notice that, for any $\boldsymbol{x} \in \mathbb{R}^d$,

$$g_\mu(\boldsymbol{x}) - g(\boldsymbol{x}) = \inf_{\boldsymbol{u}} \{g(\boldsymbol{u}) + \phi_\mu(\boldsymbol{x} - \boldsymbol{u})\} - g(\boldsymbol{x})$$
$$\geq \inf_{\boldsymbol{u}} \{\langle \mathbf{g}, \boldsymbol{u} - \boldsymbol{x} \rangle + \phi_\mu(\boldsymbol{x} - \boldsymbol{u})\}, \qquad \forall \mathbf{g} \in \partial g(\boldsymbol{x}),$$
$$= -\sup_{\boldsymbol{u}} \{\langle \mathbf{g}, \boldsymbol{x} - \boldsymbol{u} \rangle - \phi_\mu(\boldsymbol{x} - \boldsymbol{u})\}$$
$$= -\phi_\mu^\star(\mathbf{g}) = -\mu\phi^\star(\mathbf{g}), \qquad \forall \mathbf{g} \in \partial g(\boldsymbol{x}), \forall \boldsymbol{x}. \tag{39}$$

where the inequality is by the definition of subgradients. Further because we assume $dom\, \phi^\star \supseteq \cup_{\boldsymbol{x} \in \mathcal{X}} \partial g(\boldsymbol{x})$, the supermum of $\phi^\star$ is thus finite, i.e. $\|\phi^\star\|_\infty := \sup_{\mathbf{g}} \phi^\star(\mathbf{g}) < \infty$. Therefore, Eq.(39) $\geq -\mu \|\phi^\star\|_\infty$ is a valid lower bound, which completes the proof.

$\square$

## B.6 Supplementary Materials to Section 4

*Condition* B.1. The smooth function $\phi : \mathbb{R}^m \to \mathbb{R}$ in Eq.(6) should satisfy following conditions:

1. function $\phi$ is continuously differentiable, strictly convex, and $\beta'$-smooth;

2. its Fenchel conjugate function $\phi^\star(\boldsymbol{y}) := \sup_{\boldsymbol{u} \in dom\,\phi}\{\langle \boldsymbol{y}, \boldsymbol{u} \rangle - \phi(\boldsymbol{u})\}$ exists, and the domain of $\phi^\star$ is a superset of subgradients set of $g$, i.e., $dom\,\phi^\star \supseteq \cup_{\boldsymbol{x} \in \mathcal{X}} \partial g(\boldsymbol{x})$.

The first condition is common in literature. The second condition facilitates analysis through Fenchel Conjugate, which leads to various results in Lemma 4.6. While the superset requirement in the second condition is not straightforward at first glance, if $g$ is $L$-Lipschitz continuous (w.r.t. $\ell_2$ norm), we can safely replace it with a sufficient alternative that $dom\,\phi^\star \supseteq \mathcal{B}(L)$. The alternative requirement is much easier to check in practice.

There are various eligible $\phi$ function, and we list out some in Table 2. Evidently, functions in Table 2 all meet Condition B.1. However, not all $\phi$ functions in Table 2 result in an approximation function $g_{\mathsf{ME}}$ that approximates $g$ from above. The inequality in part 3 of Lemma 4.6 implies approximation

Table 2: Eligible Functions $\phi(\cdot)$ for (Generalized) Moreau Envelope

| Functions | $\phi(\boldsymbol{x})$ | dom $\phi$ | Smoothness[1] | $\phi^\star(\boldsymbol{y})$ | dom $\phi^\star$ | Convexity[1] |
|---|---|---|---|---|---|---|
| Energy[2] | $\frac{1}{p}|x|^p$ | $\mathbb{R}$ | 1 | $\frac{1}{q}|y|^q$ | $\mathbb{R}$ | 1 |
| $\ell_2$-norm | $\frac{\beta'}{2}\|\boldsymbol{x}\|_2^2$ | $\mathbb{R}^d$ | $\beta'$ | $\frac{1}{2\beta'}\|\boldsymbol{y}\|_2^2$ | $\mathbb{R}^d$ | $\frac{1}{\beta'}$ |
| Huber | $\begin{cases} \beta' \cdot \|\boldsymbol{x}\|_2^2/2, & \|\boldsymbol{x}\|_2 \leq 1, \\ \beta' \cdot \|\boldsymbol{x}\|_2 - \beta'/2, & \text{otherwise.} \end{cases}$ | $\mathbb{R}^d$ | $\beta'$ | $\frac{1}{2\beta'}\|\boldsymbol{y}\|_2^2$ | $\mathcal{B}(\beta')$ | $\frac{1}{\beta'}$ |
| Hellinger | $\beta' \cdot \sqrt{1 + \|\boldsymbol{x}\|_2^2}$ | $\mathbb{R}^d$ | $\beta'$ | $-\sqrt{\beta'^2 - \|\boldsymbol{y}\|_2^2}$ | $\mathcal{B}(\beta')$ | $\frac{1}{\beta'}$ |

[1] with respect to $l_2$-norm $\|\cdot\|_2$;  [2] $p, q > 0$ s.t. $1/p + 1/q = 1$;

from above only when $\|\phi^\star\|_\infty \leq 0$. The only eligible $\phi$ function is Hellinger. Specially, for the Hellinger function, we should set $\beta'$ to be at least $L$ in order to meet the superset requirement. The approximation function by Hellinger is drawn in Figure 4.

## B.7 Additional Experiment Results

### B.7.1 Excess generalization risks

**Linear $\ell_1$ loss with a fixed $A$.** In the main text, we reported numerical results for loss function $f(\boldsymbol{\theta}; \boldsymbol{y}, A) = \|\boldsymbol{y} - A\boldsymbol{\theta}\|_1$ where datapoints are $(\boldsymbol{y}, A)$ and $A$s' elements are random variables. Now, let us fix the matrix to be a deterministic matrix $A := \begin{bmatrix} 1 & .5 & 0 & 0 & 1 \\ .5 & .5 & 0 & 0 & 1 \\ 0 & 0 & -.5 & 0 & 1 \end{bmatrix}$. In this case, the optimal $\boldsymbol{\theta}^*$ satisfies $A\boldsymbol{\theta}^* = (-1.25, -1, -0.5)$; therefore, assumption 4.4 holds. Results are reported below in Figure 7. Similarly, in high-privacy and small-sample regimes (left bottom), C-OP outperforms other methods. One may notice the improvement is not as significant as that shown in Figure 5. We conjecture this is because the deterministic nature of $A$ weakens the advantage of C-OP that it can return a minimizer to an ERM. With a deterministic matrix $A$, SGD can also output a high-quality minimizer, even with high privacy.

**Piecewise linear loss.** We further conduct experiments under a piecewise linear function $f(\boldsymbol{\theta}; \boldsymbol{y}, A) = \max_{p \in [P]}\{\langle \boldsymbol{a}_p, \boldsymbol{y} - A\boldsymbol{\theta} \rangle + b_p\}$ whose convolution is given in Section 4.3. Experiment settings are the same as that in Figure 5, and results are shown in Figure 8. From Figure 8, we have similar observations: C-OP outperforms Noisy-SGD and Moreau in high-privacy regimes (for example, $\varepsilon = 0.2, d = 20$). But when privacy requirements become less stringent, the advantages diminish; and finally, Noisy-SGD performs the best again (for example, $\varepsilon = 5, d = 30$). This suggests that when privacy is high, C-OP might be a competitive alternative to Noisy-SGD.

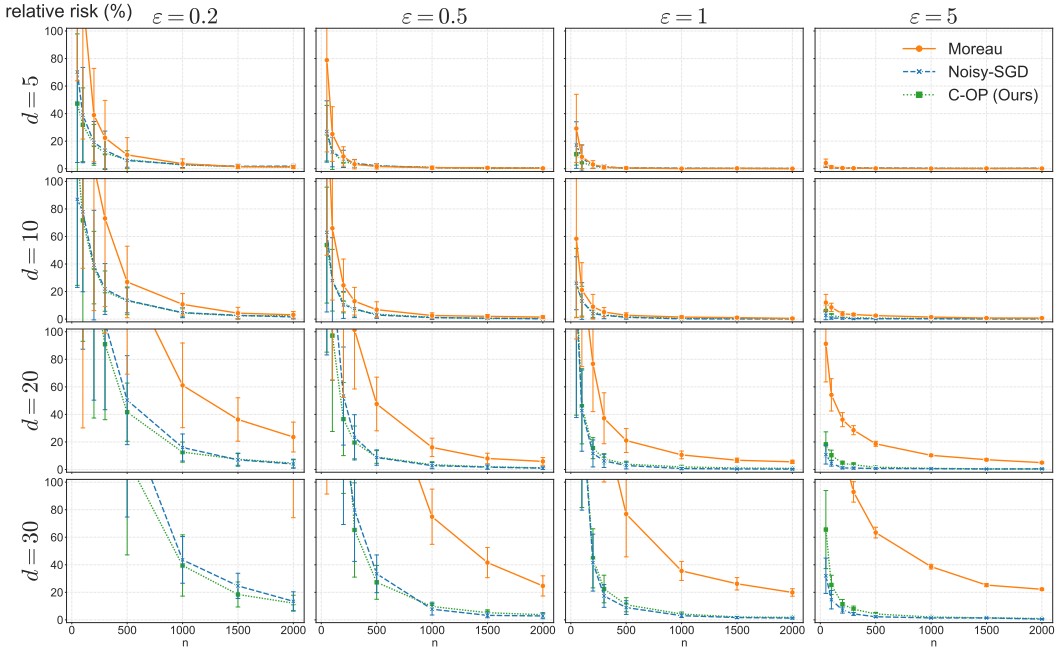

Figure 7: Relative risk v.s. sample size under $\ell_1$-norm loss. Settings are the same as in Figure 5 expect that we fix the matrix $A$ to be deterministic $A := \begin{bmatrix} 1 & .5 & 0 & 0 & 1 \\ .5 & .5 & 0 & 0 & 1 \\ 0 & 0 & -.5 & 0 & 1 \end{bmatrix}$.

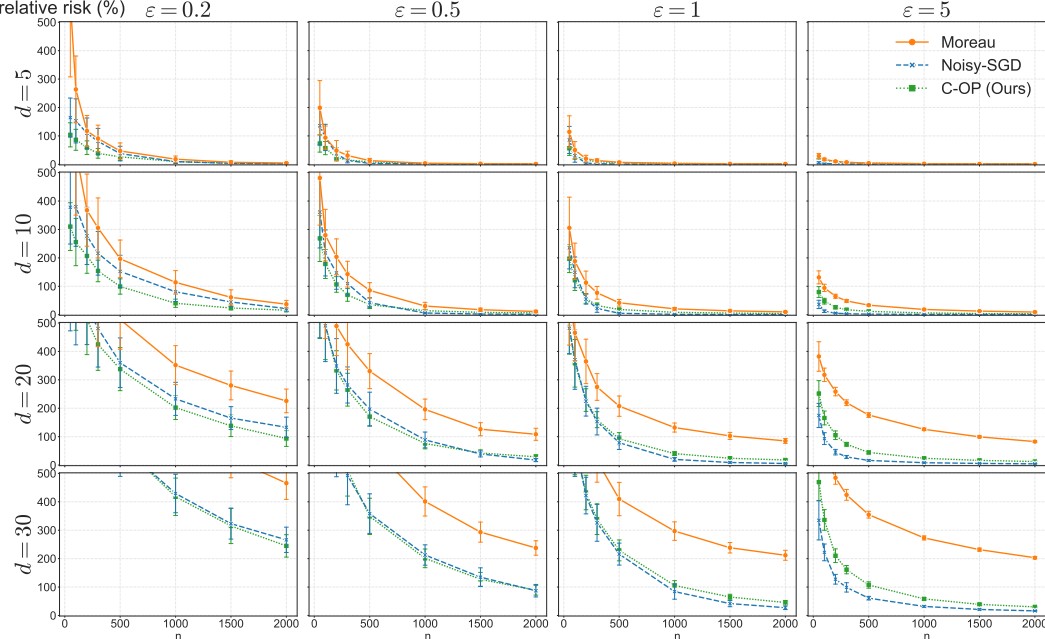

Figure 8: Relative risk v.s. sample size, under a piecewise linear loss function. Settings are the same as in Figure 5.

B.7.2 COMPUTATIONAL EFFICIENCY

For the ease of notation, let us denote $\boldsymbol{x} := A(\boldsymbol{z})\boldsymbol{\theta}$. The computational bottleneck of C-OP is the high-dimensional integration $\mathbb{E}_{\mathbf{k}}\left[\|A(\boldsymbol{z})\boldsymbol{\theta} + \mu\mathbf{k}\|_1\right] := \int_{\boldsymbol{v}\in\mathbb{R}^m} \|A(\boldsymbol{z})\boldsymbol{\theta} + \mu\boldsymbol{v}\|_1 \, \boldsymbol{k}(\boldsymbol{v}) \, d\boldsymbol{v}$ in Eq.(1). When the kernel function chosen is Gaussian kernel $\boldsymbol{k}(\boldsymbol{v}) := \Pi_{j=1}^m \left(\frac{1}{\sqrt{2\pi}} \exp\left(-\frac{v_j^2}{2}\right)\right)$, the integral admits a closed form expression:

$$\int_{\boldsymbol{v}\in\mathbb{R}^m} \|\boldsymbol{x} + \mu\boldsymbol{v}\|_1 \, \boldsymbol{k}(\boldsymbol{v}) \, d\boldsymbol{v} = \int_{v_m} \cdots \int_{v_1} \|\boldsymbol{x} + \mu\boldsymbol{v}\|_1 \, \Pi_{j=1}^m \left(\frac{1}{\sqrt{2\pi}} \exp\left(-\frac{v_j^2}{2}\right)\right) dv_1 \dots dv_m$$

$$= \|\boldsymbol{x}\|_1 + \sum_{j=1}^m \left(\mu\sqrt{2/\pi} \exp\left(-\frac{x_j^2}{2\mu^2}\right) - 2|x_j|\Phi\left(-\frac{|x_j|}{\mu}\right)\right),$$

where $\Phi(\cdot)$ is the CDF of a standard Gaussian random variable. However, for general kernels that only satisfy Condition A.1, the convolution may not admit closed-form expressions because the integral cannot be analytically done layer by layer.

We implement the integration with torchquad Gómez et al. (2021), a numerical integration module utilizing GPUs. The module conducts high-dimensional convolution by discretizing the convolution integral into small bins first, and then it calculates the function value in each bin, and lastly sums them up. Because this kind of parallelization significantly benefits from GPUs, the calculation is computationally efficient. Specifically, each integration completes within 50ms on an RTX 2060S graphics card, and the overall optimization completes within seconds. The source code is attached as supplementary materials for readers who are interested in implementation details.

Table 3: C-OP Runtime (in seconds)

| n | d | $\varepsilon = 0.2$ | $\varepsilon = 0.5$ | $\varepsilon = 1$ | $\varepsilon = 5$ |
|---|---|---|---|---|---|
| 100 | 5 | 0.49 | 0.53 | 0.54 | 0.51 |
| | 10 | 0.70 | 0.83 | 0.88 | 0.83 |
| | 20 | 2.73 | 3.11 | 3.48 | 3.16 |
| | 30 | 4.02 | 5.19 | 6.16 | 5.85 |
| 500 | 5 | 0.76 | 0.91 | 0.67 | 0.71 |
| | 10 | 0.90 | 0.98 | 1.00 | 1.01 |
| | 20 | 4.41 | 4.02 | 3.94 | 3.71 |
| | 30 | 9.24 | 8.96 | 6.63 | 6.03 |
| 1000 | 5 | 0.98 | 1.08 | 0.89 | 0.90 |
| | 10 | 1.16 | 1.28 | 1.35 | 1.36 |
| | 20 | 5.03 | 5.16 | 5.42 | 5.28 |
| | 30 | 12.04 | 10.23 | 8.98 | 8.64 |
| 2000 | 5 | 1.58 | 1.52 | 1.35 | 1.36 |
| | 10 | 1.84 | 2.09 | 2.01 | 2.04 |
| | 20 | 7.05 | 7.72 | 8.11 | 7.51 |
| | 30 | 14.75 | 13.88 | 12.17 | 13.41 |

