# OpenReview forum: "Exploiting Hidden Symmetry to Improve Objective Perturbation for DP Linear Learners with a Nonsmooth L1-Norm"
_ICLR.cc/2025/Conference — ICLR 2025 Poster_

### Official Review · Reviewer_cGwQ · 2024-10-29

**Soundness:** 3
**Presentation:** 3
**Contribution:** 2
**Rating:** 6
**Confidence:** 3

**Summary:**

The paper studies differentially-private optimization of nonsmooth convex functions. A subclass of such functions with "hidden $\ell_1$ structure" is defined, for which a smoothing+objective perturbation is shown to achieve the optimal rate (which is not the case in general).
Some experiments show the algorithm preforms well compared to baselines.

**Strengths:**

The paper studies an important problem, and does a good job motivating it with an excellent introduction, posing it compared to prior work.

There are several technical contributions throughout the paper, in order to apply the OP approach and prove its utility. These are explained fairly well in my opinion, and are generally of general interest.

The experiments make a convincing point that the approach considered in this work might have practical importance.

**Weaknesses:**

The assumptions made throughout the paper, the formulation of which are arguably the main contribution, seem pretty limiting.

Two things particularly seem limiting:

1. Although the word "hidden" structure appears in the title and in the name of the main assumption, it actually **needs to be known explicitly** by the algorithm, if I am not mistaken. The algorithm seems to need to know the decomposition into the "$\ell_1$-component" and the rest in the first place, which only holds for very structured problems such as GLMs or computing quantiles (which is indeed the one considered in the experiment).

2. The results neglect the dependence on the matrix dimension $m$, which is considered a small constant. This is what really allows the smoothing approach to work here, which would otherwise have a strong dimension dependence (preventing the optimal rate, as far as I can tell). While this is true for some structured problems considered in the paper, this seems pretty specific.

**Questions:**

I would appreciate the author's comments on the two issues I raised in the previous "weakness" box.

Some additional minor comments:

- Line 076: It is unclear at this point of the paper what $\theta$ vs. $z$ are.
- Line 082: "and etc." should be simply "etc.".
- Several places, including Lines 092 and 107, use \cite (or \citet) where \citep is more appropriate.
- Line 130: $B(R)$ is the ball around the origin, I suppose? This is unclear.
- If possible, rather slightly move Algorithm 1 so that it appears at the beginning or end of the paragraph it currently cuts-off in the middle.
- Assumption 4.4 and related discussion: the notation $\|\cdot\|_{-\infty}$ is used, perhaps should be $\infty$ instead of $-\infty$? Otherwise I don't understand this notation.

---

> ### Author Response · Authors · 2024-11-18
> **Response by the authors**
>
> Dear Reviewer cGwQ,
>
> We sincerely thank you for your time reading our work and for providing insightful comments. We have revised the manuscript according to your suggestions. Below, we will respond to your concerns one by one.
>
> **Weakness 1. The structure needs to be known explicitly and it holds only for very structure problems.**
>
> **Response:** Thank you very much for the comment. Indeed, the structure of "$\ell_1$+smooth" needs to be known explicitly. Because only in this way can we know where to apply convolution. While we have tried to consider general settings, it turns out the most interesting model under which we can clearly illustrate the benefits of convolution is the "$\ell_1$+smooth". So, we focus our discussion mainly on this model.
>
> However, that does not mean our method can only be applied to models with the said structure. In fact, our method can be applied to some other problems that do not follow the assumed structure, such as piecewise linear function and soft-thresholding function in Section 4.3. These extensions demonstrate the potential in convolution on other nonsmooth problems beyond GLMs and quantiles. While we understand these extensions are again very specific, they suggest that our idea could have broader impacts beyond the "$\ell_1$+smooth" structure.
>
> As you may notice, in all considered examples ($\ell_1$-regularized GLMs, quantiles, piecewise linear, etc), there exists nonsmooth terms in the form of $\max(0,x)$ or $|x|$ implicitly. So, you are right that we need to decompose (or reformulate) the loss function into the "$\ell_1$+smooth" structure before using our method. That's the exact reason why we call it a "hidden" structure: the $\ell_1$ part is hidden behind. That being said, we agree with you that the word "hidden" is somewhat misleading. It may mistakenly remind people of hidden variables or hidden layers. Considering this, we decide to remove the word "hidden" from the title, and refined wording pertinent to it.
>
> As part of responses to your this concern, we have revised the introduction to the assumed structure to highlight (i) the $\ell_1$ part is implicit, instead of something hidden, and (ii) a decomposition (or reformulation) might be necessary. With that, we hope your concern is addressed. Once again, thank you for this comment.
>
> **Weakness 2. The matrix dimension $m$ is assumed to be a small constant, which seems pretty specific.**
>
> **Response:** Thank you! This is a very sharp observation. Indeed, the dimension $m$ should be a small constant; otherwise, there will be a dimension dependence, leading to a slower rate.
>
> This is indeed a limitation, but it is not unique to our work. It is an inherent limitation of OP. Simply put, any terms interacting with $\theta$ must output variables in a lower dimensional space; otherwise the dimensionality issue carries over to convergence rates, preventing it from being optimal. The limitation dates back to the first work on OP (Kifer et al. 2012, COLT), and is frequently mentioned in recent studies (see Appendix G in Redberg et al. 2023, NeurIPS; footnote on page 2 & Lemma 21 in Agarwal et al. 2023, COLT). We clearly notice this limitation, and thus assume $m$ to be a small constant. Nevertheless, comparing to the preceding works that all consider GLMs with $m=1$, our work allowing $m$ to be constants other than 1 actually advances the technique.
>
> Though we did not emphasize the dimension $m$ because addressing the dimensionality issue is not the focus of our work, we have taken extra care with it when developing theoretical results. For instance, we explicitly present the impact of $m$ in all main theorems (i.e. we leave $m$ there explicitly, rather than hiding it with symbols $\mathcal{O}(\cdot)$ or $\lesssim$). We believe doing so can let readers see the impact of $m$ clearly; thus, they can determine if our method fits their needs. Nevertheless, given the concern raised by you, we concur there is a necessity to highlight this limitation to the community. Therefore, we add a paragraph "limitation of our work" to the Conclusion section, and clearly state the dimensionality issue on $m$. With that, we hope our response addresses your concerns.
>
> **Minor comments on notation and typo**
>
> **Response:** Thank your for careful reading. we have fixed these minor issues accordingly.
> - $\mathbf{z}$ and $\theta$ are now properly introduced in line 76
> - Thank you! We have fixed it.
> - Thank you! We have fixed \citep issues in lines 92 and 107, among others.
> - Yes, $B(R)$ is the ball around the origin. Now, the definition is introduced.
> - Thank you for this! We have fixed it.
> - Thank you for pointing out this! The notation $||\mathbf{x}||_{-\infty}:=\min(|x_1|, |x_2|, \dots, |x_m|)$ denotes the minimal absolute value of elements of $\mathbf{x}$. It is properly introduced now.
>
> Once again, thank you for all the comments! We are more than happy to take follow-up questions.
>
> Yours,
>
> Authors of Submission 2949

---

> > ### Comment · Reviewer_cGwQ · 2024-11-26
> > **Response to rebuttal**
> >
> > I thank the authors for their detailed response. I choose to maintain my score.

---

### Official Review · Reviewer_SBAs · 2024-11-04

**Soundness:** 3
**Presentation:** 3
**Contribution:** 3
**Rating:** 6
**Confidence:** 2

**Summary:**

This work proposes C-OP, an algorithm for differential privacy. C-OP is designed for a specific class of problems, namely one with a smooth component and a non-smooth component containing the $\ell_1$ norm. The key design is the random perturbation within the $\ell_1$ norm. The algorithm achieves $\epsilon-\delta$ privacy with optimal excess risk.

**Strengths:**

1. The theoretical analysis is technically impactful. Lemma 4.3 as a technical lemma is very useful.
2. The presentation follows good logic.
3. The proposed algorithm is effective in experiments.

**Weaknesses:**

1. It would be better if the authors could discuss the explicit result of the convolution. The reviewer did not check it, but they believe that the $\mathbb{E}_k[A(z)\theta+\mu k]$ can be explicitly written in all choices of $k$ of interest. If not, the authors are expected to discuss how the convolution is implemented.
2. It would be interesting to see the intuition of applying convolution to the part of $\ell_1$ loss only instead of the entire objective function. See Question 1.
3. In several subfigures of Figure 5, the plot goes out of the boundary.
4. The discussion about the choice of $\mu$ and $\sigma$ is not detailed enough. It would be better if the authors could discuss whether $\mu$ and $\sigma$ should increase or decrease with the change of other hyperparameters.
5. According to my understanding, the algorithm cannot be applied to the smooth setting. See also Question 2.

**Questions:**

1. As the reviewer is not quite familiar with the literature, it would be interesting to see why previous methods that try to use convolutional smoothing fail to achieve the optimal risk but the algorithm in this paper can. Is it due to the new analysis techniques in this paper? If not, what consequence does the convolution on the smooth component bring about?
2. As the scale of $A(z)$ decays, the problem becomes close to a smooth problem, and it seems that more noise needs to be injected into the convolution to preserve privacy. How can the authors explain this using Theorem 3?

---

> ### Author Response · Authors · 2024-11-18
> **Response by the authors (1/2)**
>
> Dear Reviewer SBAs,
>
> We sincerely thank you for reviewing our work and providing valuable comments. We got inspired a lot by your comments and revised the manuscript accordingly. Below, we will respond to your concerns one by one. You may also check the revised manuscript for these revisions highlighted in blue.
>
> **weakness 1: explicit results of the convolution.**
>
> **Response:** Thank you for the question. Generally speaking, the convolution $E_{\mathbf{k}}[||\mathbf{x}+\mu \mathbf{k}||_1]$ cannot be explicitly written out, because it involves a high-dimensional integral. But under some coordinate-separable kernels, such as Gaussian kernel, it indeed admits a closed-form expression, $\mathbb{E}{[||\mathbf{x}+\mu\mathbf{k}||_1]}=||\mathbf{x}||_1  + \sum_j\left(\mu \cdot \sqrt{2/\pi} \cdot exp(-\frac{x_j^2}{2\mu^2}) -2|x_j|\Phi(-\frac{|x_j|}{\mu})]\right)$. Despite the special case, we hope to keep our technical results as general as possible: you may notice that many technical lemmas in Section 3 & 4 are stated for all kernels that only meet a weaker Condition A.1. For the reason of generality, we did not explicitly write out closed-form expressions for all considered kernels.
>
> Nevertheless, we agree with you that we could do a better job explaining how we implemented the convolution in experiments. Simply put, the convolution was numerically calculated with the python package *torchquad*, which is a integration module for high-dimensional integral utilizing GPUs. We had a short discussion on the implementation in Appendix 7.2 in the initial manuscript. But after noticing the concern from you, we believe we should provide more details for readers who are interested in implementations. So we added more discussion in the revised manuscript (highlighted in blue, see Appendix 7.2), and also uploaded the source code as supplementary materials. You may find how we implemented the convolution in the method *ObjectivePert.F\_beta\_l1\_GPU* in *models.Objective\_Pert*. Thank you very much.
>
> **Weakness 2 & Question 1: Intuition about why previous methods with convolution fail but we succeed**
>
> **Response:** Thanks for this question, we are happy to take this question because you actually have noticed the answers already (they are in your question). Indeed, it is because (i) we only smooth out the nonsmooth part (which brings minimal approximation erros) and (ii) we devloped more involved analyais exploiting the symmetry of $\ell_1$-norm. Let us explain them one by one.
>
> - (i) we only smoothed out the nonsmooth $\ell_1$ part. Smoothing always comes with approximation errors, even if you are smoothing a smooth function. When doing smoothing, the philosophy is that we hope to gain smoothness at minimal costs. So, we only need to smooth the $\ell_1$ part. If we smooth the entire function, there will be extra errors from the smooth part. For example, let us smooth out a simple function $f(x)=|x|+\frac{1}{2}x^2$ with Gaussian kernel and $\mu=1$. The table below compares approximation errors under the two cases: only smoothing the nonsmooth part $|x|$, and smoothing the entire function. From the table, it is clear that smoothing the entire function brings extra errors, and the additional errors will further harm convergence. So, when doing smoothing, we only need to focus on the nonsmooth part.
> | approx. error      | x = -2 | -1.5  | -1    | -0.5  | 0     | 0.5   | 1     | 1.5   | 2     |
> | ------------------ | ------ | ----- | ----- | ----- | ----- | ----- | ----- | ----- | ----- |
> | smooth$\|x\|$ only | 0.016  | 0.058 | 0.166 | 0.395 | 0.797 | 0.395 | 0.166 | 0.058 | 0.016 |
> | smooth entire$f$   | 0.516  | 0.558 | 0.666 | 0.895 | 1.297 | 0.895 | 0.666 | 0.558 | 0.516 |
>
> - (ii) we developed more involved analysis. Existing works all use the uniform approximation error for convergence analysis. But our analysis shows that the uniform approximation error is too conservative (see Figure 1). So we characterized pointwise error by exploiting the symmetry of $\ell_1$ (Lemma 4.1 - 4.3), and use this tighter error for analysis. In other words, previous works thought the cost of gaining smoothness is too high to achieve optimal rates. But our analysis shows that the cost is not as high as previously thought, and it actually does not harm convergence rates.
>
> **Weakness 3: visualization-related issue**.
>
> **Response:** Thanks for pointing out this. We present the figures in this way intentionally. This is because all subfigures in Figure 5 share y-axis. Sharing y-axis allows us to see the impact of $\varepsilon$. But because $\varepsilon$ affects performance a lot, the results under $\varepsilon=0.2$ and $\varepsilon=5$ differ significantly. Thus, we had to set y-lim to some interval to ensure curves in all subfigures recognizable (especially for subfigures under $\varepsilon=5$). Nevertheless, we will think about a better way to present the figures. Thank you for bringing this visualization-related issue to us!

---

> ### Author Response · Authors · 2024-11-18
> **Response by the authors (2/2)**
>
> **Weakness 4: elaborate choices of $\mu$ and $\sigma$**.
>
> **Response:** Thank your very much for this suggestion. We agree that the choice of $\mu$ is not clear enough: it only appears once in alg 1 step 1. Given the important role of $\mu$, we believe it is necessary to illustrate how we choose $\mu$ and how it is affected by other parameters. In short, $\mu$ is derived from an equation (alg 1 step 1), and it depends on the kernel function chosen, privacy parameters, and sample size. Specifically,
>
> - *the heavier the tail of kernel function is, the smaller the value of $\mu$ should be*. This aligns with our technical results: recall that lemmas in Section 4 indicate that a smaller $\mu$ implies better approximations. And in theory, a kernel with heavier tails indeed leads to better approximations.
> - *larger $(\varepsilon,\delta)$ lead to a decreasing $\mu$*. This relationship is also intuitive. When $(\varepsilon,\delta)$ go up, we become less care about privacy; so we can use an approximation function that is closer to the original (nonsmooth) function (aka a one that approximates better). Eventually, when $\varepsilon=\infty$ or $\delta=1$, we take $\mu=0$, i.e. we use the original nonsmooth function, and don't worry about privacy at all.
> - *larger sample size $n$ implies smaller $\mu$*. This is again intuitive. As sample size goes up, each datapoint's impact on $\hat{\theta}$ becomes less significant. So we can use a better approximated function (ie, a smaller $\mu$).
>
> With the discussion here, we hope your concerns regarding the value of $\mu$ are addressed. As part of responses, we revised the manuscript accordingly. Now, you can find more discussions on the choice of $\mu$ after Lemma 3.3.
>
> The choice of $\sigma$ depends only on the size of data space and boundedness parameter of $A(\cdot)$. A larger data space and a higher boundedness parameter both lead to a higher $\sigma$. This follows a large body of literature on OP; so we believe the current discussion and results in Theorem 3.2 should be sufficient.
>
> **Weakness 5 & Question 2: apply to smooth settings?**.
>
> **Response:** Thanks for this question! Short answer: our method can be applied to smooth settings, but this is not recommended. Let us explain it.
>
>  Here is a quick background: the classic OP method is applicable to smooth settings (see Kifer et al. 2012 or Bassily et al 2019). But it cannot be applied to nonsmooth settings. Our work is trying to make OP applicable to a specific class of nonsmooth loss functions. Now, let us come back to your concern: *the algorithm cannot be applied to the smooth setting*. In fact, if the loss function at hand is smooth, we can invoke classic OP directly, and no need to use our method. While one can forcibly use our method and apply convolution to a smooth problem, it will lead to extra approximation errors as shown above. Essentially, if the problem is already smooth, there is no incentive to do so.
>
> As for Question 2, when $A(z)$ decays, the boundedness parameter $\bar{A}$ decreases. So Theorem 3.2 suggests less noise (a smaller $\sigma$). This aligns with our intuition: when $A(z)$ decays, the problem indeed becomes close to a smooth problem. Because smoothness implies a certain kind of stability of the minimizer (i.e. minimizers are less sensitive to any single datapoint), less noise is enough to hide privacy.
>
> (**PS**. After reading question 2, we feel there might be a misunderstanding regarding our algorithm. We would like to provide an explanation here just in case. The noise for privacy is not injected into the convolution; instead, the noise $\mathbf{b}$, appearing in a linear term $\frac{\<\mathbf{b},\mathbf{\theta}\>}{n}$, is injected into the entire loss function, see Step 3 & 4 in alg 1. Intuitively, the convolution only serves as a pre-processing step, and it has nothing to do with the noise injection. The noise is injected into the smoothed function derived from the convolution, not into the convolution smoothing process itself.)
>
>
> **In sum**,  once again, we sincerely thank you for taking the time to review our work and for providing valuable suggestions. We hope our response addresses your concerns. If you have any follow-up questions, please feel free to let us know. We are more than happy to provide further explanations. If your concerns are well-addressed, we sincerely request a re-assessment of our work, and please consider to raise your rating. Thank you!
>
> Yours,
>
> Authors of Submission 2949

---

> > ### Comment · Reviewer_SBAs · 2024-11-18
> >
> > I would like to thank the authors for the detailed explanation. All my concerns are resolved, and it is great to see the updated version of the paper. I have updated my scoring.

---

> > > ### Author Response · Authors · 2024-11-19
> > >
> > > Thank you very much for raising your rating. Once again, thank you for all the comments that helped us improve the manuscript significantly.

---

### Official Review · Reviewer_4oV9 · 2024-11-04

**Soundness:** 4
**Presentation:** 4
**Contribution:** 3
**Rating:** 6
**Confidence:** 3

**Summary:**

The authors extend the Objective Perturbation (OP) framework for differentially private (DP) convex optimization, traditionally applied to smooth loss functions, to accommodate a specific class of nonsmooth convex losses. Specifically, they address nonsmooth functions characterized by an implicit $\ell_1$-norm structure. Their method involves smoothing only the nonsmooth component of the objective function via convolution, followed by the application of an OP-based algorithm to uphold DP guarantees.

**Strengths:**

The problem is well-motivated, and the core ideas of the paper are clearly articulated. Although the use of convolution to obtain a smooth approximation of a nonsmooth function is not novel in itself, it enables the derivation of interesting and novel theoretical results. The primary innovation lies in leveraging an approximation function that provides an upper bound on the original nonsmooth objective. This approach results in a new decomposition of the excess generalization risk, which is subsequently used to derive optimal generalization rates.

**Weaknesses:**

In my opinion, the main limitation of the paper lies in the applicability of Assumption 4.4. For example: the $\ell_1$-norm structured nonsmooth functions which is the focus of this paper is often utilized in sparse linear regression where $\theta^*$ is sparse. The  Assumption 4.4 does not seem to hold for such cases. Besides, without the knowledge of $\theta^*$, one cannot verify this assumption easily.

**Questions:**

Can the authors comment on the applicability and verifiability of Assumption 4.4? It would strengthen the paper to provide some concrete examples related to real-world applications where the proposed assumption holds.

---

> ### Author Response · Authors · 2024-11-18
> **Response by the authors**
>
> Dear Reviewer 4oV9,
>
> We sincerely thank you for your time reading our paper and for providing insightful comments.
>
> A quick answer to your concern regarding the applicability and verifiability of assumption 4.4 is that, generally speaking, it is indeed hard to verify the assumption in practice. As you already pointed out, this is simply because verifying the assumption requires the knowledge of $\theta^*$; but in practice, $\theta^*$ is known to nobody. Moreover, for many problems, the closed-form expression of $\theta^*$ may not analytically exist, making it even harder to verify the assumption.
>
> Fortunately, there are many important real-world applications where the assumption naturally holds, such as quantile regression, problems with loss function $\max(0, x)$, and $\ell_1$ regularized regression (e.g. LASSO, under some conditions). We explain these examples below, and believe that the insights drawn from these examples can be extended to more general cases.
>
> - **Quantile Regression**. The $r$-th quantile loss is $f(\mathbf{\theta}, \mathbf{z}:=(y,\mathbf{x}))= |y - \mathbf{x}^\top \mathbf{\theta}|/2 + (r-1/2)(y - \mathbf{x}^\top\mathbf{\theta})$. In this case, $A(\mathbf{z})=(y, -\mathbf{x}^\top)/2$. Suppose data $(y,\mathbf{x})$ follows a linear model $y=\left<\mathbf{x},\mathbf{\theta}\right>+\epsilon$ with $x_1\equiv 1$ being the intercept term and $\epsilon$ being the noise with a CDF $\Phi$. Then $\mathbf{\theta}^*= \mathbf{\theta} + [\Phi_r^{-1}, 0, \dots, 0]^\top$ where $\Phi_r^{-1}$ is the $r$-th quantile of the noise. So $|A(\mathbf{z})\mathbf{\theta}^*| = |y - \mathbf{x}^\top\mathbf{\theta}^*|/2= |-\Phi_r^{-1}|/2$. Then, as long as the $r$-th quantile of $\epsilon$ is not 0 (i.e. $\Phi_r^{-1}\neq 0$), we can choose $\tau:=|\Phi_r^{-1}|/2$ to make assumption 4.4 true. The only assumption here is the $r$-th quantile of $\epsilon$ is not 0. This is very mild and similar to identifiability assumption in learning theory. So, for quantile regression, we can safely say Assumption 4.4 always holds. (BTW, arguments naturally extend to high-dimensional cases.)
>
> - **ReLU**. Let $f(x) = \max(0, x)$. Equivalently, $f(x) = |x|/2+ x / 2$. This form takes the same structure as an $r$-th quantile loss. So taking $r\rightarrow 1$, we know the assumption holds.
>
> - **LASSO** (conditions apply). Consider the lasso problem $\theta^*:=\arg\min_{\theta}\lambda |\theta|_{1} + 1/2 E[(y-\mathbf{x}^\top\theta)^2]$.  We know $\theta^*$ has a closed form expression $\theta^*:=\mathsf{sgn}(\theta^\star)(\theta^\star-\lambda)^+$ with $\theta^\star$ being the minimizer to the problem without the regularizer. In this case $A(\mathbf{z})\theta^*=\theta^*$. Whether it is sparse or not depends on (i) the non-regularized minimizer $\mathbf{\theta}^\star$ and (ii) the coefficient $\lambda$. If $\mathbf{\theta}^\star$ is not sparse and $\lambda$ is a small constant, then $\mathbf{\theta}^*$ is also not sparse; thus assumption 4.4 holds. Back to the weakness you mentioned. Indeed, if $\theta^\star$ is assumed to be sparse, then $\theta^*$ is also sparse and assumption 4.4 does not hold. But again in practice, nobody knows $\theta^\star$. Often, what we do is to choose a value of $\lambda$ for some extent of sparsity rather than full sparsity. If so, assumption 4.4 might hold.
>
> These examples are common problems in reality, and also play a role as building blocks for other interesting problems. With these examples, we believe assumption 4.4. is sufficiently mild.
>
> We want to thank you for bringing the concrete example of sparse linear regression (e.g. LASSO discussed above) to our attention, because when thinking about the assumption, we find a very interesting trade-off: if we require high sparsity (a larger $\lambda$), the assumption will be violated, so we cannot enjoy the benefits of convolution, leading to a slower convergence rate and a higher generalization risk. But on the other hand, if we require low sparsity (a smaller $\lambda$), the assumption holds and we can enjoy convolution; however, not being sparse also leads to a higher risk because the underlying model is assumed to be sparse. It seems there is a trade-off between not being sparse to enjoy benefits of convolution and being sparse to enjoy model alignments. Nevertheless, this is beyond the scope of our work, and we would like to explore it in the future (as it is a really interesting trade-off).
>
> In sum, while assumption 4.4 is hard to verify in practice, it holds for some applications. We have updated the manuscript to include these examples as part of responses to your concern. You may check out the updated manuscript for the revision highlighted in blue. Thank you again for your time and valuable comments. Hope our response addresses your concerns. Please feel free to let us know if you have any other questions, and we are more than happy to take follow-up questions.
>
>
> Yours,
>
> Authors of Submission 2949

---

> > ### Comment · Reviewer_4oV9 · 2024-11-26
> >
> > I appreciate the authors’ effort in providing examples, which help address some of my concerns. However, the verifiability of Assumption 4.4 and applicability to Lasso (with sparse $\theta^*$) remain unresolved. After considering the response, I have decided to maintain my score.

---

### Meta-Review · Area_Chair_coVe · 2024-12-19

**Metareview:**

The paper considers differentially private convex optimization, aiming to address problems with nonsmooth objective functions. When the loss function is smooth, a standard approach known as the objective perturbation is effective and well-understood. However, much less is known for the nonsmooth cases. The paper considers specific, structured, nonsmooth problems that can be decomposed into smooth + nonsmooth $\ell_1$ terms. The proposed approach is to apply convolution to smoothen the $\ell_1$ part and then apply the objective perturbation. The paper then proves that, under certain assumptions such as the dimension being much larger than the number of data points, this approach provably works. Although somewhat limited due to the required strong assumptions, the paper was ultimately judged to be sufficiently novel to merit acceptance.

**Additional Comments On Reviewer Discussion:**

The main concerns raised in the reviews were about the required assumptions for the proposed approach to work. These were clarified by the authors in the rebuttal phase. Some of the assumptions (such as, e.g., low dimensionality $m$) seem in line with prior work, while the assumption about explicit knowledge of the structured nonsmooth component limits the work to applications such as GLMs and quantile regression. Nevertheless, given how little is known about the nonsmooth cases despite these problems being the focus on recent research, the stated applications seem appropriate for a conference paper.

---

### Decision · Program_Chairs · 2025-01-22

Accept (Poster)